ecology

agricultural intensification, avian declines, ecological restoration, land-use change, life-history traits, woodland birds

**Author for correspondence:**
Glen C. Bain
e-mail: glen.bain@utas.edu.au

# Changing bird communities of an agricultural landscape: declines in arboreal foragers, increases in large species

Glen C. Bain[1], Michael A. MacDonald[2], Rowena Hamer[3], Riana Gardiner[3], Chris N. Johnson[3] and Menna E. Jones[3]

[1]School of Natural Sciences, University of Tasmania, Private Bag 55, Hobart, Tasmania 7005, Australia
[2]RSPB Centre for Conservation Science, RSPB Cymru, Castlebridge 3, 5-19 Cowbridge Road East, Cardiff CF11 9AB, UK
[3]School of Natural Sciences, University of Tasmania, Hobart, Tasmania 7005, Australia

GCB, 0000-0003-1760-6706; MAM, 0000-0002-1061-8504; RH, 0000-0002-9063-5426; CNJ, 0000-0002-9719-3771

Birds are declining in agricultural landscapes around the world. The causes of these declines can be better understood by analysing change in groups of species that share life-history traits. We investigated how land-use change has affected birds of the Tasmanian Midlands, one of Australia's oldest agricultural landscapes and a focus of habitat restoration. We surveyed birds at 72 sites, some of which were previously surveyed in 1996–1998, and tested relationships of current patterns of abundance and community composition to landscape and patch-level environmental characteristics. Fourth-corner modelling showed strong negative responses of aerial foragers and exotics to increasing woodland cover; arboreal foragers were positively associated with projective foliage cover; and small-bodied species were reduced by the presence of a hyperaggressive species of native honeyeater, the noisy miner (*Manorina melanocephala*). Analysis of change suggests increases in large-bodied granivorous or carnivorous birds and declines in some arboreal foragers and nectarivores. Changes in species richness were best explained by changes in noisy miner abundance and levels of surrounding woodland cover. We encourage restoration practitioners to trial novel planting configurations that may confer resistance to invasion by noisy miners, and a continued long-term monitoring effort to reveal the effects of future land-use change on Tasmanian birds.

## 1. Introduction

Agricultural intensification is a major cause of global biodiversity loss [1]. Beyond the direct clearing and fragmentation of habitat

that typically accompanies agriculture, farm management practices can modify wildlife communities through their effects on the abundance of food, shelter and predators. For example, rising levels of pesticide use to protect crops from herbivory have had wide-ranging negative impacts on invertebrate populations [2] and have subsequently been implicated as a major contributor in the decline of farmland bird populations in Europe and North America [3–5]. By contrast, corvids and mammalian mesopredators (for example, feral cats *Felis catus* and raccoons *Procyon lotor*) are thought to have benefitted from an increased food supply on farms and fewer large predators [6,7]. The mechanization of agriculture and expansion of irrigated lands also results in the removal of discrete habitat elements such as paddock trees, surface rocks, coarse woody debris and hedgerows [8,9]. These features are often the only source of cover for animals in farm landscapes and are used by a range of taxa for breeding, thermoregulation and as sources of food [10–12].

Understanding how land use affects wildlife communities is essential if we are to predict their response to future environmental change, whether that results from continued agricultural intensification or landscape restoration. This is especially important given that the agricultural sectors need to meet a projected increase in global food demand of between 25 and 70% [13]. Land-use change associated with agriculture can affect species differently according to their life-history traits such as body size, diet or dispersal ability. Relationships between functional traits of species and population change can offer insight into the mechanisms by which land use affects wildlife [14,15]. This understanding can be applied to produce more targeted conservation strategies and may allow for broader approaches that protect several species at once [16,17]. Functional traits can also be used to estimate the contribution of species to ecosystem services and, therefore, how those services might respond to land-use change [18].

Birds have long been used as indicators of environmental condition and can be effective indicator species for the conservation of other threatened vertebrates [19]. They occupy many ecological niches, respond relatively rapidly to changes in habitat, are straightforward to monitor, and perform important ecological functions such as seed dispersal, pest control and pollination [20]. We surveyed birds in the Tasmanian Midlands, one of Australia's oldest agricultural landscapes, a National Biodiversity Hotspot and the focus of an ambitious habitat restoration programme [21]. The bird community of the Midlands has high biodiversity value, with 10 Tasmanian endemic species and several distinct subspecies (electronic supplementary material, table S1). Tasmania, more generally, is important for many species of terrestrial bird (at least 19) that migrate to the island from mainland Australia to breed each year [22,23].

An increasing threat to local bird communities is secondary clearing of remnant woodlands for the installation of large pivot irrigation systems [24]. Agricultural fragmentation in the Midlands has also led to the dominance of an aggressive species of native honeyeater, the noisy miner (*Manorina melanocephala*), in many small patches of habitat [25]. Noisy miners are native to Australia's eastern coast but have become overabundant [26]. The noisy miner's exclusion of small birds from suitable woodland habitat is listed as a *Key Threatening Process* under federal environment legislation [27]. The productive landscapes of the Midlands also support very high densities of feral cats which are a major predatory threat to birdlife [28], although it is unknown how mortality from cat predation might translate to population viability of birds.

Our first aim was to assess changes in the Midlands bird community over the past 20 years as a consequence of land-use change. The only broad-scale survey of birds in the Tasmanian Midlands was previously conducted in 1996–1998 (hereafter the 1997 survey period, [25]). We repeated surveys at 34 historical study sites in order to explore the effects of habitat clearing, revegetation and changes in noisy miner abundance on bird communities. Changes in bird abundance between survey periods allowed for the identification of target species and functional groups of birds to be either encouraged or discouraged by restoration efforts.

Second, we explored whether the response of birds to land-use change is trait-mediated. Previous research has identified ground-foraging insectivores as particularly vulnerable to habitat degradation and predation by introduced species in Australia [29]. Nectarivores and large-bodied birds were also found to have declined in agricultural landscapes of New South Wales [15]. Unfortunately, Tasmania lacks informative terrestrial bird survey data when compared to other parts of Australia. This is particularly the case in the Midlands, which is mostly privately owned. To address this gap and provide context to the effects of land-use change, we described current patterns of bird abundance and community composition and how these are influenced by environmental factors. We use the results of our study to make practical recommendations for restoring habitat for terrestrial birds in Tasmania.

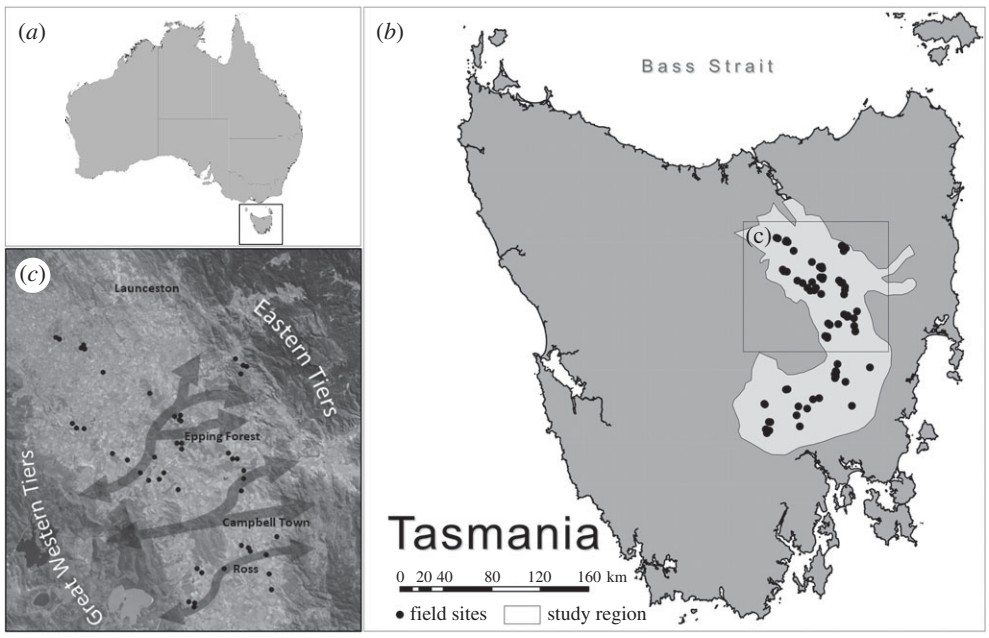

**Figure 1.** Location of survey sites in the Midlands of Tasmania, Australia. Inset (c) shows a satellite image of part of the study region, which is largely cleared of native vegetation. Arrows signify the conceptual east–west connections of the Midlands restoration programme where revegetation is planned to occur.

## 2. Methods

### 2.1. Study region

Tasmania lies at the southern-most tip of Australia and is separated from the Australian mainland by Bass Strait (250 km, figure 1). Islands of the Furneaux Group and King Island act as stopovers for birds that migrate to Tasmania in the spring and summer [22,23]. The Midlands Biodiversity Hotspot (*ca* 8000 km²) is defined by a low annual rainfall (less than 600 mm) and lies within north (Macquarie and Esk Rivers) and south (Jordan, Clyde and Coal Rivers) flowing catchments that are bordered by the mountainous Eastern and Western Tiers of Tasmania. Prior to European settlement, the Midlands consisted of grassy or heathy temperate eucalypt woodlands and native grasslands. More than 200 years of clearing for agriculture has left less than 10% of native woodland and less than 3% of native grasslands remaining [21]. The region is now predominately exotic pastures for livestock grazing (primarily sheep), and crops of cereals, oilseeds and other high-value yields such as poppies for the pharmaceutical industry. Agricultural intensification continues to result in deforestation and particularly the loss of scattered paddock trees [24].

Temperatures range from mean daily maximums of 21.9°C in summer and 11.4°C in winter to mean minimum temperatures of 8.8°C in summer and 1.9°C in winter. However, Tasmania experienced several extreme weather events during our study, including a period of drought in 2015, Tasmania's coldest winter in 50 years with abnormal snowfalls across the Midlands (2015), followed by the warmest summer since records began (2016–2017) and record levels of rainfall that caused significant flooding at some of our field sites [30]. Such extreme weather events, especially heatwaves, are expected to become more frequent in the Midlands under the effects of climate change [31].

### 2.2. Survey sites

We surveyed 72 sites representing the range of habitat types in the Midlands. This included 52 woodland remnants and five small (less than 6 ha) mixed-species eucalypt plantings, ranging in age since planting of approximately 20–30 years. To document birds that were using the agricultural matrix, we also conducted 2 ha/20 min bird surveys [32] in five native grasslands, five pasture sites and five areas dominated by the exotic weed, gorse (*Ulex europaeus*). These sites were placed *a priori* and usually included at least one paddock tree.

Remnant woodlands reflected a range of patch sizes and levels of modification: 15 small (0–20 hectares), 25 medium (20–200 hectares) and 12 large (greater than 200 hectares). The dominant tree species (canopy 10–30 m) were black peppermint (*Eucalyptus amygdalina*), cabbage gum (*E. pauciflora*), white gum (*E. viminalis*), silver peppermint (*E. tenuiramis*) and gum-topped stringybark (*E. delegatensis*). Typically, small and degraded patches of woodland were characterized by exotic pasture grasses throughout their interior, bracken fern (*Pteridium esculentum*) and introduced weeds including gorse and hawthorn (*Crataegus monogyna*). The mid-storey of these patches was generally sparse.

Larger and less disturbed woodlands were more structurally complex with a greater diversity of low branching trees including wattles (*Acacia dealbata, A. mearnsii, A. axillaris*), sheoaks (*Allocasuarina verticillata, A. littoralis*), native cherry, (*Exocarpos cupressiformis*), yellow bottlebrush (*Callistemon pallidus*) and silver banksia (*Banksia marginata*). Dense thickets of prickly box (*Bursaria spinosa*) were common at some sites. The understorey of intact woodlands comprised of a patchy mosaic of heathy groundcovers (*Lissanthe strigosa, Epacris impressa*), bracken fern, native perennial grasses (*Themeda triandra, Poa labillardieri, Austrostipa* spp.), sedges (*Lepidosperma* spp.) and rushes (*Juncus* spp.). Swards of spiny-head mat-rush (*Lomandra longifolia*) were a prominent feature at many sites.

## 2.3. Bird surveys

Surveys were conducted by a single observer (G.C.B.) between May 2015 and February 2017 both at dawn and dusk (89% of surveys within three hours of sunrise). Surveys were undertaken blind as to the occupancy of species at field sites during the 1997 survey period. Each site was surveyed in clear weather on up to six occasions: twice during the non-breeding season (March–August) and twice each in spring (September–October) and summer (November–February). Due to limited farm access and flooding, not all sites could be surveyed six times. All sites were, however, surveyed in both winter and spring/summer to ensure that seasonal migrants were accounted for.

To enable a comparison with historical bird data, we repeated the survey methods of MacDonald & Kirkpatrick [25]. A 100 m wide fixed transect was placed parallel to the longest axis of the woodland and crossed from one side to the other. For larger woodlands, transects began at one edge and extended 800 m into the interior. These were placed randomly if their location had not already been established by historical surveys. Each transect was further divided into 200 m segments. Transects were walked at a constant pace with each segment surveyed for 10 min. All species detected by sight or sound were recorded, noting in which segment they occurred and their number. Birds that were heard but not detected on the transect were recorded as *offsite* and were not included in our final analysis. Historical surveys were also completed by a single observer (M.M.). One original site was excluded from all analyses of change in the bird community because it had become invaded by gorse such that it was no longer possible to walk a transect that was comparable to earlier surveys.

## 2.4. Environmental data

At the centre of woodland sites and plantings, we established two intersecting $10 \times 50$ m plots and recorded the following site-scale attributes: the number of alive and dead stems (greater than 10 cm diameter at breast height, DBH), the DBH of alive and dead stems (cm) from which basal area was later calculated (m$^2$), average canopy height (m) and the summative length of fallen logs (greater than 10 cm in diameter, m). For analysis, these data were extrapolated beyond the sampling plot to be expressed per hectare. Plots were positioned on survey transects in the centre of woodlands and plantings to accommodate concurrent research on habitat use by mammals [28,33]. A line-point intercept method ($2 \times 50$ m transects) was used to estimate the per cent cover of ground substrates which were classified into eight categories: leaf litter, bare earth, rocks, forbs and other low herbaceous plants, shrubs (greater than 30 cm in height, including bracken fern), sedges, mosses and lichen. Shannon's diversity index (H′) was calculated to represent the heterogeneity of groundcover [34].

We used the software program ArcMap (v. 10.4.1, [35]) to calculate the size (ha) and shape complexity of woodland remnants. The shape was calculated as the corrected perimeter to area ratio using the formula shape $=$ perimeter of woodland$/\sqrt{4\pi(\text{area of woodland})}$. Satellite data and the geospatial layer, TASVEG 3.0, were used to classify landcover within a buffer zone of 1 km radius that was centred on the middle of bird survey transects. TASVEG is a state government digital map that depicts the extent of 162 vegetation communities in Tasmania [36]. We updated the spatial layer to more accurately reflect vegetation composition at our survey sites, for example, where recent land clearing had occurred. For historical survey sites, patch size and landcover were also determined from

**Table 1.** Summary of landcover types within 1 km of the centre of bird survey transects at woodland ($n = 52$) and planting sites ($n = 5$).

| landcover class | mean % (s.d.) | min (%) | max (%) |
| --- | --- | --- | --- |
| eucalypt woodlands | 35.3 (26.6) | 0.2 | 96.5 |
| non-eucalypt woodlands | 0.8 (0.4) | 0.0 | 18.9 |
| eucalypt plantation | 0.4 (2.7) | 0.0 | 20.7 |
| pine plantation | 0.4 (0.3) | 0.0 | 17.0 |
| production agriculture | 54.8 (3.8) | 0.0 | 99.1 |
| native grassland | 6.3 (1.6) | 0.0 | 56.5 |
| open water | 2.2 (0.6) | 0.0 | 26.2 |

digitized aerial images collected in 1997 to allow calculation of change between the 1997 and 2017 surveys. Seven categories of landcover were identified: (1) native eucalypt woodlands, including mixed eucalypt plantings; (2) native non-eucalypt woodlands; (3) productive eucalypt plantations; (4) pine (*Pinus radiata*) plantations; (5) agricultural pastures and other exotic vegetation; (6) open water (mostly farm dams); and (7) native grasslands (table 1). Categories 1–4 were further classified as 'woody vegetation'. Cover of native grasslands could not be determined from aerial imagery, so for all analyses, the distribution of grasslands in 1997 was assumed to be unchanged from that in 2017. Shannon's Diversity Index was again used to represent heterogeneity of landcover.

We counted the number of centre-pivot irrigation systems present within 1 km of survey sites as a proxy for land-use intensity. Pivot irrigators need relatively flat terrain to operate and usually require large areas of land (up to 300 ha) to be entirely cleared of native vegetation. Climate data (mean annual rainfall, rainfall seasonality, mean annual temperature), as well as the elevation of survey sites and projective cover of woody foliage (foliage projective cover, FPC) were derived from geospatial information systems [37,38]. Values for FPC were averaged over the area of each bird survey transect. Finally, the density of noisy miners at survey sites was considered as an environmental variable because previous research demonstrated strong effects of noisy miners on the abundance of other bird species [39]. Density of miners was determined from bird surveys as described above (see §2.3).

## 2.5. Trait data

All birds observed were categorized according to species traits (electronic supplementary material, tables S1–S3) using data extracted from the *Handbook of Australian, New Zealand and Antarctic Birds* [40–46]. These traits were body size, diet, foraging height, native status and movement pattern (table 2). We acknowledge that many species do not exclusively fit one category type for each trait, but for the purposes of analysis, birds were categorized according to their primary behaviours.

## 2.6. Statistical analyses

Our analysis was performed in five stages. First, we used the package *BORAL* in R to produce a model-based unconstrained ordination of bird-count data for the visualization of species–site relationships [47,48]. In all analyses of multivariate data, we pooled observations of birds from transect surveys at each site and specified a negative binomial distribution. We included a fixed row effect when generating our ordination to account for differences in the total abundance of birds at survey sites such that the resulting ordination reflects species composition.

Next, we used the package *mvabund* (v. 3.13.1) to test for effects of environmental variables on bird community composition at survey sites [49]. Candidate environmental variables were reduced to a final set of 10 (table 3) by generating a correlation matrix (Spearman's rank) and eliminating those variables that demonstrated high levels of collinearity ($r_s > 0.6$). Woody vegetation cover and distance to the nearest patch of woodland ($r_s = 0.71$) were associated; only woody vegetation cover was retained in our models. Elevation and FPC were considered as representative of climatic variables (e.g. mean temperature, rainfall seasonality) because these too were highly correlated.

*Mvabund* implements a generalized linear model (GLM) framework to analyse multivariate abundance data and fits a separate GLM to each species. This technique accounts better for the mean–

**Table 2.** Summary of trait variables used to explain variation across bird species in their response to environmental variables.

| trait | category | example species |
|-------|----------|-----------------|
| body size | very large (>1000 g) | wedge-tailed eagle, Australian shelduck |
| | large (100–1000 g) | sulphur-crested cockatoo, grey currawong |
| | medium (25–100 g) | eastern rosella, fan-tailed cuckoo |
| | small (<25 g) | scarlet robin, superb fairy-wren |
| diet | invertebrates | Australian magpie, welcome swallow |
| | vertebrates | brown falcon, nankeen kestrel |
| | seeds and grains | galah, house sparrow |
| | nectar | musk lorikeet, eastern spinebill |
| | plants | grey teal, Australian shelduck |
| | generalists | black currawong, forest raven |
| foraging height | terrestrial | flame robin, yellow-rumped thornbill |
| | arboreal | striated pardalote, little wattlebird |
| | arboreal/terrestrial | grey-shrike thrush, grey butcherbird |
| | aerial | welcome swallow, brown falcon |
| movement pattern | resident/sedentary | yellow-throated honeyeater, grey butcherbird |
| | migratory[a] | silvereye, dusky woodswallow |
| | nomadic[b] | crescent honeyeater, musk lorikeet |
| native status | native to Tasmania | green rosella, spotted pardalote |
| | exotic to Australia | common starling, European goldfinch |
| | introduced to Tasmania | laughing kookaburra, little corella |

[a]Migratory species included only those birds known to migrate to mainland Australia annually, including partial migrants.
[b]Nomadic species also included altitudinal migrants and species described as dispersive in the literature.

variance relationship of count data than do distance-based multivariate analyses (e.g. canonical correspondence analysis) and allows formal tests of bird responses to environmental variables at both the community and species-specific levels [49]. An offset (area of transect in hectares log-transformed) was used in our model to adjust for differences in the area surveyed at each site and the effort argument was included to account for variation in the number of times a site was surveyed. Wald test statistics and $p$-values were determined from 999 resampling iterations via the PIT-trap method. Adjusted univariate $p$-values were calculated for individual species to test for their response to environmental variables.

We then used the *traitglm* function in *mvabund* to evaluate how bird species' traits influence their relative abundance and response to environmental variables [50]. This method also uses an extension of a GLM, fitting a single predictive model to all species across all sites simultaneously. Three matrices of environmental data, species-abundance data and species-trait data were used to calculate a fourth matrix of trait–environment interaction coefficients, or 'fourth-corner' terms [51]. For visual interpretation, we generated a heat-map of the standardized coefficients and used the LASSO penalty to remove all interactions that failed to improve model fit [51]. We also used *traitglm* to model bird count data without specifying a trait matrix. This method effectively fits a multivariate species distribution model and assumes a different environmental response for each species. *Mvabund* does not yet support offsets when modelling fourth-corner problems and using LASSO in the model fitting process. While the trait analysis does not, therefore, account for differences in the area of each transect surveyed, the results remain informative. Observed traits were unrelated to woodland patch size and so this limitation is expected to have minimal influence.

In our fourth analysis, we used GLMs to assess which elements of landscape change best explained differences in species richness at historical survey sites between the 1997 and 2017 survey periods. Here, species richness was the total number of native bird species recorded at a site over the duration of each survey period. A Poisson distribution of errors is usually assumed in models of discrete count data, but in this case, the response variable could be both positive and negative (species richness may have increased

**Table 3.** Summary of a multivariate analysis (*manyGLM*) testing for the effects of environmental variables on bird community composition. The *p*-values (less than 0.05 italics) and Wald statistics are given for the effect of variables at the community level. Estimates ± s.e. are for individual species that contributed significantly to the variance in community composition. The sign of the estimate (positive or negative) indicates the direction of a species response to the environmental variable.

| environmental variable | Wald (1 d.f.) | community *p*-value | species where *p* < 0.05 | estimate ± s.e. |
|---|---|---|---|---|
| woody vegetation cover | 20.95 | *0.001* | Australian magpie (*Cracticus tibicen*) | −0.015 ± 0.007 |
| | | | common starling (*Sturnus vulgaris*) | −0.029 ± 0.006 |
| | | | eastern rosella (*Platycercus eximius*) | −0.031 ± 0.012 |
| | | | European goldfinch (*Carduelis carduelis*) | −0.037 ± 0.008 |
| | | | forest raven (*Corvus tasmanicus*) | −0.010 ± 0.005 |
| | | | grey butcherbird (*Cracticus torquatus*) | −0.019 ± 0.006 |
| | | | black-headed honeyeater (*Melithreptus affinis*) | +0.053 ± 0.048 |
| | | | crescent honeyeater (*Phylidonyris pyrrhopterus*) | +0.034 ± 0.013 |
| | | | eastern spinebill (*Acanthorhynchus tenuirostris*) | +0.095 ± 0.033 |
| | | | fan-tailed cuckoo (*Cacomantis flabelliformis*) | +0.058 ± 0.034 |
| | | | grey currawong (*Strepera versicolor*) | +0.039 ± 0.013 |
| | | | grey shrike-thrush (*Colluricincla harmonica*) | +0.064 ± 0.017 |
| | | | scarlet robin (*Petroica boodang*) | +0.020 ± 0.006 |
| | | | yellow wattlebird (*Anthochaera paradoxa*) | +0.044 ± 0.008 |
| | | | yellow-throated honeyeater (*Lichenostomus flavicollis*) | +0.013 ± 0.010 |
| foliage projective cover | 13.59 | *0.011* | — | — |
| miner density | 21.03 | *0.012* | brown thornbill (*Acanthiza pusilla*) | −1.033 ± 0.189 |
| | | | European goldfinch (*Carduelis carduelis*) | −1.394 ± 0.238 |
| | | | grey fantail (*Rhipidura albiscapa*) | −0.952 ± 0.179 |
| | | | scarlet robin (*Petroica boodang*) | −1.481 ± 0.378 |
| | | | silvereye (*Zosterops lateralis*) | −3.096 ± 0.683 |
| | | | superb fairy-wren (*Malurus cyaneus*) | −1.033 ± 0.222 |
| | | | yellow-rumped thornbill (*Acanthiza chrysorrhoa*) | −2.590 ± 0.504 |
| | | | Australian magpie (*Cracticus tibicen*) | +0.865 ± 0.129 |
| | | | eastern rosella (*Platycercus eximius*) | +1.738 ± 0.217 |
| landcover diversity H′ | 11.56 | *0.013* | — | — |
| elevation | 13.65 | *0.014* | grey fantail (*Rhipidura albiscapa*) | −0.006 ± 0.001 |
| leaf litter cover | 10.61 | *0.039* | — | — |
| basal area | 9.95 | 0.088 | — | — |
| shape complexity | 12.23 | 0.118 | — | — |
| canopy height | 9.57 | 0.468 | — | — |
| groundcover diversity H′ | 8.74 | 0.546 | — | — |

or declined). After examination of diagnostic plots, we found that a Gaussian distribution with identity link provided a good fit to our data. In these models, we included combinations of the following explanatory variables: patch size (ha in 1997), change in patch size (ha), per cent change in the amount of woody vegetation cover within 1 km, change in noisy miner density (miners ha$^{-1}$) and the number of pivot irrigation systems within 1 km in 2017. We used an information-theoretic approach to assess model performance and ranked models by Akaike's information criterion corrected for a small sample size (AICc, [52]).

Finally, we used the log-response ratio (lnRR) to assess changes in species densities at historical survey sites. This was calculated as $\ln(\bar{x}^1/\bar{x}^2)$, where $\bar{x}^1$ was the mean density of a species in the 2017 survey period across all sites where that species was present (in any survey year) and $\bar{x}^2$ was the equivalent but for the 1997 survey period. Bird species with a low absolute density might still have a high lnRR because this measure reflects proportionate change. The lnRR was also used to assess changes in the density of birds that shared species traits. Here, $\bar{x}^1$ and $\bar{x}^2$ were the mean densities of all species present in a trait category in the 2017 and 1997 survey periods respectively. Mean ± s.e. is reported where appropriate.

# 3. Results

A total of 91 species was recorded across all our bird surveys (electronic supplementary material, tables S1–S3). Of these, 72 were recorded during transect surveys, including five species that are exotic to Australia and three that have been introduced to Tasmania from the Australian mainland. Seven additional species were detected and identified from calls offsite, eight more were observed during concurrent 2 ha/20 min bird surveys in woodlands (not analysed in this study, electronic supplementary material, table S2) and four were found only in either pastures or grasslands (electronic supplementary material, table S3). These additional records tended to be of birds that were rare (e.g. peregrine falcon *Falco peregrinus*), flying overhead (e.g. silver gull *Chroicocephalus novaehollandiae*), more typical of open habitats (e.g. striated fieldwren *Calamanthus fuliginosus*, banded lapwing *Vanellus tricolor*), or of domesticated birds and feral species with large aural detection distances (e.g. helmeted guineafowl *Numida meleagris*, Indian peafowl *Pavo cristatus*).

The size and type of habitat patch surveyed was related to community composition and species richness of birds. Species richness of native birds was highest for large remnant woodlands (23.09 ± 1.07), followed by medium (19.75 ± 1.35) and small remnants (14.47 ± 1.60), areas of gorse (12.5 ± 0.5), native grasslands (11.5 ± 1.77), plantings (10.6 ± 1.08) and pastures (7.25 ± 1.03). Because of the influence of noisy miners and a greater range in patch size, woodlands classed as medium size varied greatly in bird community composition (figure 2). Some medium-sized remnants supported species that were typical of large intact woodlands, while others were most similar in species composition to small and degraded remnants. Thus, when species were pooled across sites, medium remnants were the most species-rich (58 species, $n = 25$) followed by large (52, $n = 12$) and small remnants (48, $n = 15$), native grasslands (31), plantings (25), gorse (23) and pastures (20).

A variety of bird species used non-woodland habitats and some small patches of woodland had very high bird densities. Many of the birds that we recorded in pastures, grasslands and gorse were observed flying between nearby patches of woodland and used paddock trees within our survey plots as 'stepping stones'. Some species were more frequently recorded in these habitat types. For example, white-fronted chats (*Epthianura albifrons*) were most often seen in areas of gorse, and tree martins (*Petrochelidon nigricans*) were consistently recorded in those grassland and pasture sites where large paddock trees were present. Bird densities (birds ha$^{-1}$, pooling species) were higher in small remnants (8.70 ± 0.87) and plantings (7.01 ± 1.56) than in medium (6.70 ± 0.70) and large remnants (5.04 ± 1.00, $X^2 = 8.25$, d.f. = 3, $p = 0.04$). This pattern was largely the result of a higher abundance of small-bodied and often introduced species (e.g. greenfinch *Chloris chloris*, house sparrow *Passer domesticus*, and common starling *Sturnus vulgaris*) in small remnants and plantings.

## 3.1. Relationships between environmental variables and the bird community

Bird-community composition was significantly influenced by 6 of the 10 environmental variables that we examined (table 3). Woody vegetation cover had the strongest effect, followed by FPC, density of noisy miners, landcover diversity, elevation and the per cent cover of leaf litter. Fifteen species responded significantly ($p < 0.05$) to increasing cover of woody vegetation. Of these, six responded negatively and nine species, including three that are endemic to Tasmania, responded positively (table 3). Univariate

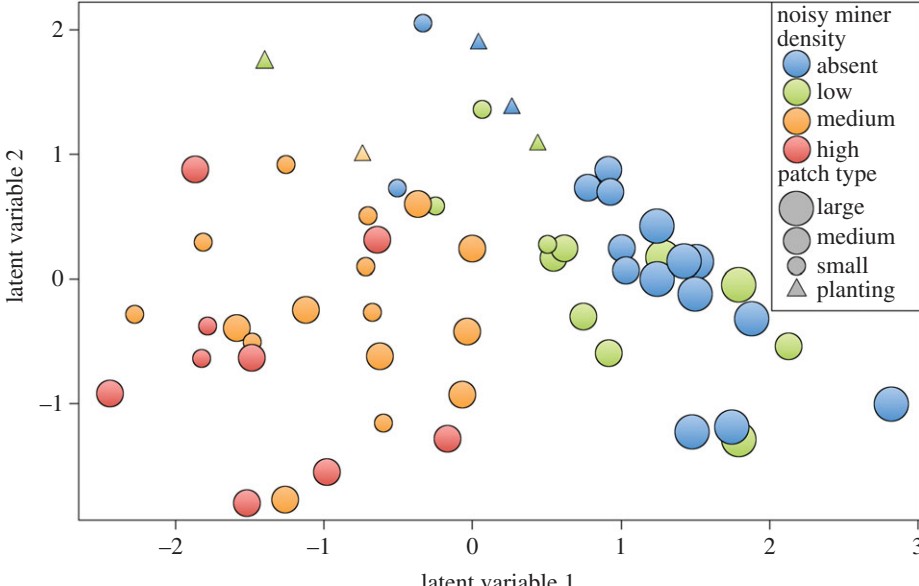

**Figure 2.** An unconstrained ordination of the bird community at woodland remnants (circles) and planting sites (triangles). Colours signify low (less than 0.6), medium (0.6–2) and high (greater than 2) noisy miner densities (miners ha$^{-1}$). Large (greater than 200 ha), medium (20–200 ha) and small (less than 20 ha) patches are indicated by the size of data points.

tests showed that no individual species contributed significantly to the effects of FPC and the cover of leaf litter on the multivariate bird community. Increasing density of noisy miners negatively affected seven small-bodied birds (less than 25 g) but was positively associated with two species, the Australian magpie (*Cracticus tibicen*) and eastern rosella (*Platycercus eximius*). Despite elevation having a strong effect at the community level, the grey fantail (*Rhipidura albiscapa*) was the only species to respond significantly negatively in univariate tests (table 3), although black-headed honeyeaters (*Melithreptus affinis*) tended to be more common in woodlands at higher elevations ($p = 0.072$).

Life-history traits of birds explained their response to environmental variables. The negative influence of noisy miners on small birds was once again highlighted by our trait analysis (figure 3). Large birds showed no response to miner density but were more common at lower elevations and at sites with a greater diversity of surrounding landcover types. The positive association between miners and birds introduced to Tasmania reflects a higher abundance of little and long-billed corellas (*Cacatua sanguinea*, *Cacatua tenuirostris*) and laughing kookaburras (*Dacelo novaeguineae*) in miner-dominated woodlands (figure 3). These three species have been introduced to Tasmania from elsewhere in Australia. However, species exotic to Australia were negatively associated with miners. Exotic species also showed a strong negative response to increasing cover of woody vegetation and were more common where the basal area of tree stands was high.

The diet and foraging habits of birds also moderated their response to environmental variables (figure 3). Arboreal foragers were more abundant at sites with greater FPC, more leaf litter, fewer miners and at higher elevations. Aerial foragers were negatively associated with woody vegetation cover and FPC but preferred sites with a simplified groundcover and diverse composition of surrounding landcover. Birds with a mainly granivorous diet were more common in woodlands at lower elevations and where FPC and woody vegetation cover were low. In contrast, generalists and nectarivores were more abundant with increasing cover of woody vegetation.

Our species distribution model indicates more complex relationships between environmental variables and individual species (figure 4). For example, sulphur-crested cockatoos (*Cacatua galerita*) were more likely to be found in areas of high woodland cover but also preferred higher landcover diversity, higher groundcover diversity and woodlands with less leaf litter. This apparently inconsistent relationship could reflect the movement of cockatoos between large woodland sites where they were frequently observed during the day and smaller satellite patches of degraded woodland where they foraged in the morning and evening. Other parrot species (e.g. blue-winged parrot *Neophema chrysostoma*, galah *Eolophus roseicapillus*, eastern rosella) were also more common in areas with diverse landcover but small-bodied and exotic birds (house sparrow, greenfinch and European goldfinch) had the opposite response.

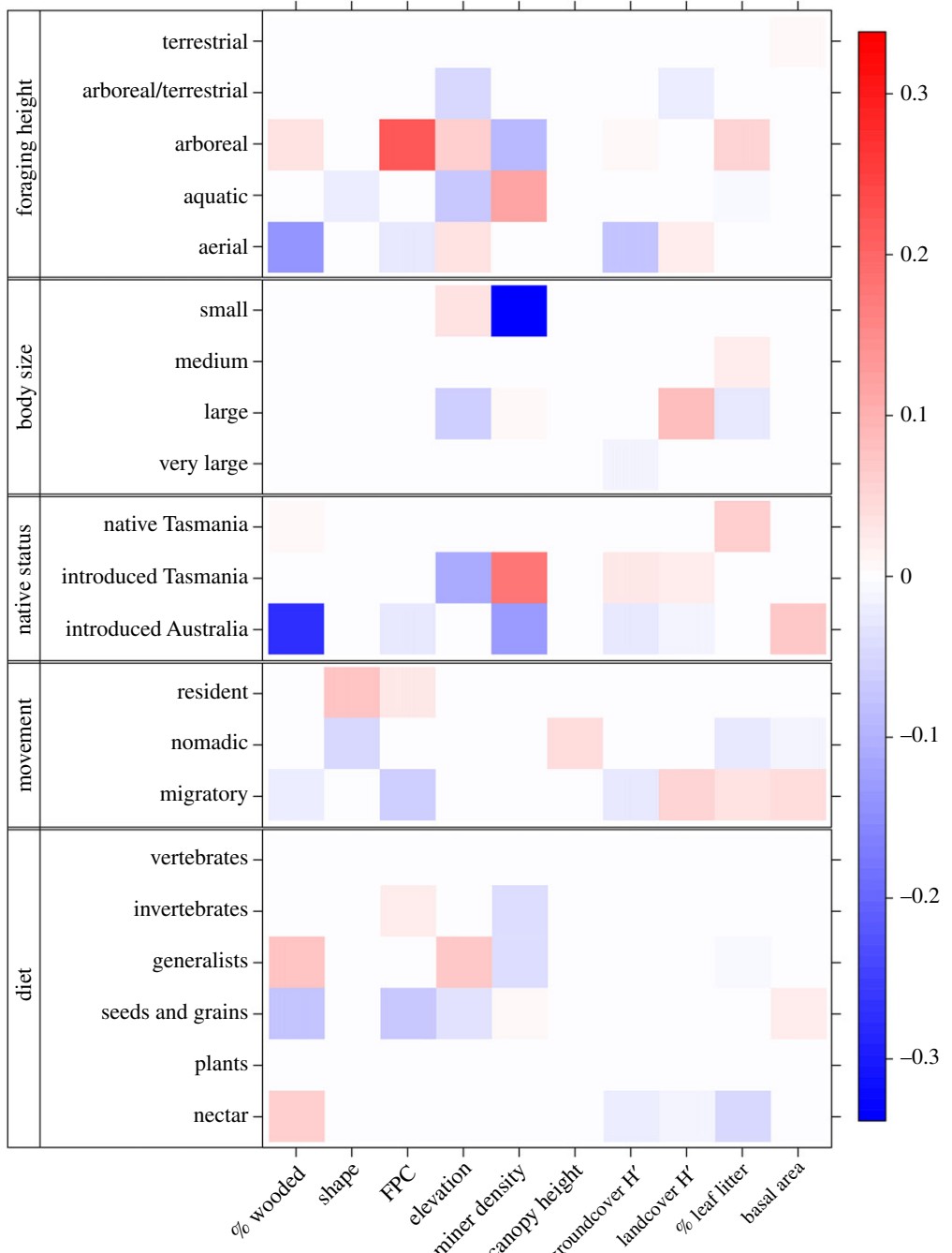

**Figure 3.** The relationship between birds that share species traits and environmental variables. Colours represent the strength of interactions (shading) and their direction (blue, negative and red, positive). The scale bar indicates the values of fourth-corner coefficients.

## 3.2. Models of change in species richness

Changes in species richness at historical survey sites were related to changes in woodland cover and densities of noisy miners. Overall, though, there was generally little change in native species richness (−1.48 ± 1.2, range = 0–18) with nearly half of all sites (15/33) gaining or losing fewer than two species (electronic supplementary material, figure S1). The most parsimonious models (Δ AICc < 7 units, [52]) explaining these changes always included the effects of change in noisy miner density (table 4). Sites where noisy miners had increased in number were more likely to have experienced a decline in native species richness. Change in woody vegetation cover was also a significant predictor: sites that had a decline in surrounding woody vegetation also experienced a decline in species richness. The initial size of the woodland patch had a smaller effect on change in species richness, but larger patches tended to have fewer species recorded than previously (electronic supplementary material, figure S2).

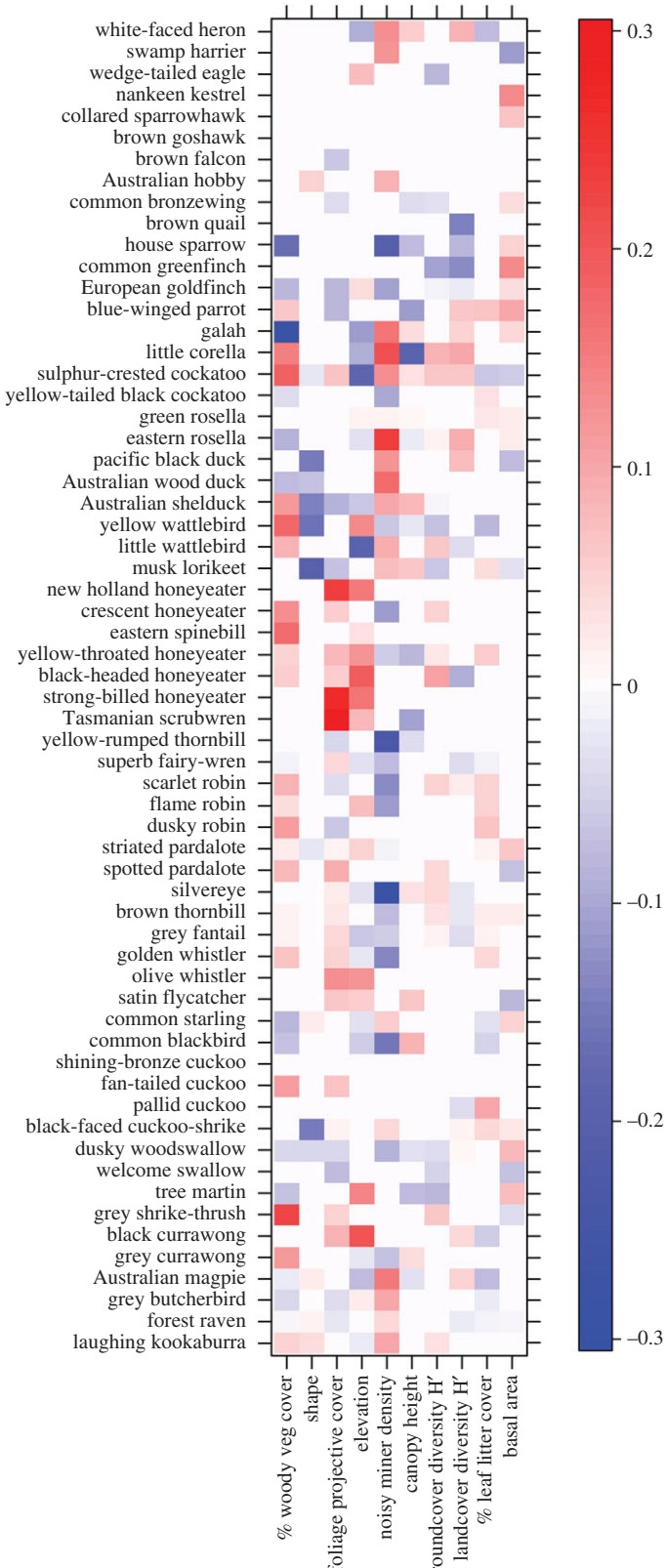

**Figure 4.** Relationships between bird species abundance and environmental variables. Species are ordered according to diet. Colours represent the strength of interactions (shading) and their direction (blue, negative and red, positive). The scale bar indicates the values of fourth-corner coefficients.

The best model included just these three predictors and explained 45% (adj. $R^2$) of the variation in the net change in species richness. The top three models, however, were all within two Δ AICc units and carried 38%, 23% and 16% of the weight, respectively (table 4).

**Table 4.** Regression analysis of the relationship between the net change in native species richness at historical survey sites ($n = 33$) and changes in noisy miner density, woody vegetation cover and patch size. Also included as covariates were the initial size of the woodland patch surveyed and the number of pivot irrigation systems present within 1 km of the survey site in 2017. Models are ranked by AICc. See electronic supplementary material, table S5 for models with $W < 0.10$.

| Δ native species richness | | | | | | | |
|---|---|---|---|---|---|---|---|
| AICc | Δ AICc | W | Δ noisy miner density | Δ woody vegetation cover | patch size | Δ patch size | pivot irrigators |
| 210.36 | 0.00 | 0.38 | −5.456 ± 1.369 | 0.378 ± 0.180 | 0.004 ± 0.002 | — | — |
| 211.34 | 0.98 | 0.23 | −5.286 ± 1.422 | 0.380 ± 0.187 | — | — | — |
| 212.07 | 1.71 | 0.16 | −5.857 ± 1.417 | 0.424 ± 0.185 | 0.004 ± 0.002 | 0.247 ± 0.233 | — |
| 212.97 | 2.61 | 0.10 | −5.941 ± 1.452 | — | — | — | — |

Only three survey sites experienced a reduction in patch size from clearing. Thus, our analysis may not have had sufficient power to detect any effects of change in patch size on species richness. The number of pivot irrigation systems increased dramatically between 1997 and 2017, from three irrigators within 1 km of two survey sites, to 33 irrigators at 12 survey sites. Yet, there was no relationship between the number of nearby irrigation systems and changes in species richness.

Change in densities of noisy miners was highly variable between sites (range = −1.30 to +1.43 ha$^{-1}$) but the mean density of noisy miners at historical study sites was lower in the 2017 survey period (−0.24 ± 0.12 ha$^{-1}$). Noisy miners could be a candidate for observer variation between survey periods, but we think this is unlikely given the species is easily detectable. Miners were recorded at 31/33 sites in 1997 and at 28/33 sites in 2017. Four woodlands had miners present in 1997 but no miners in 2017, including one site where miner density was previously as high as 1.21 miners ha$^{-1}$. Miners were recorded at only one site where they were formerly absent and at a very low density (0.02 ha$^{-1}$).

Woody vegetation cover had changed at 14/33 sites. At 12 of these, woodland cover had declined with an average loss of 3% (range = 1 to 8%). Native woodland was replaced by agricultural pastures and exotic vegetation at 11 sites and by both open water (farm dam) and pasture at one location. Two small remnants experienced an increase in surrounding woody vegetation cover due to the establishment of a pine plantation near one site (+17%) and a eucalypt plantation at the other (+21%). In the woodland remnant adjacent to the eucalypt plantation, native species richness more than doubled from 12 to 30 species (three additional species were heard offsite). Noisy miner densities at this site were also one-third of what they were previously (1.86 versus 0.56 miners ha$^{-1}$). Only a narrow road separated one edge of the remnant from the eucalypt plantation and several birds were observed crossing between the two habitat types. The result was a community that combined many species more typical of large woodland remnants (e.g. pallid cuckoo *Cacomantis pallidus*, scarlet robin *Petroica boodang*) as well as those common to small sites (e.g. noisy miners and grey butcherbirds *Cracticus torquatus*). In fact, the bird community appeared to change between the first segment (0–200 m) of the transect, which was nearest to the plantation (25 species observed), and the second segment (200–400 m, 13 species). By contrast, the pine plantation had no effect on species richness, which was the same (10 species) in the nearby woodland remnant for both survey periods.

At the other extreme, native species richness more than halved at one medium-sized remnant (88 ha) from 31 species to just 13. The reasons for this change are not immediately clear, but noisy miner density at the site was much higher in 2017 than had previously been recorded (0.56 versus 1.75 miners ha$^{-1}$). Landcover at this site was unchanged but the property manager had begun using a small area in the centre of the woodland for disposing of farm waste.

## 3.3. Changes in abundance

Birds associated with water (e.g. swamp harrier *Circus approximans*, white-faced heron *Egretta novaehollandiae*, Australian shelduck *Tadorna tadornoides*) were more abundant in the 2017 survey period (figure 5). This was also reflected in the lnRR of birds with a herbivorous diet and very large body size, which were mostly species of ducks (figure 6). Granivores and raptors also appeared to be more common. Swamp harriers had the highest proportionate increase among species for which lnRR was calculated, followed by galahs (*Eolophus roseicapillus*) and the little wattlebird (*Anthochaera chrysoptera*), a species common in urban environments of Tasmania. Sulphur-crested cockatoos and the introduced

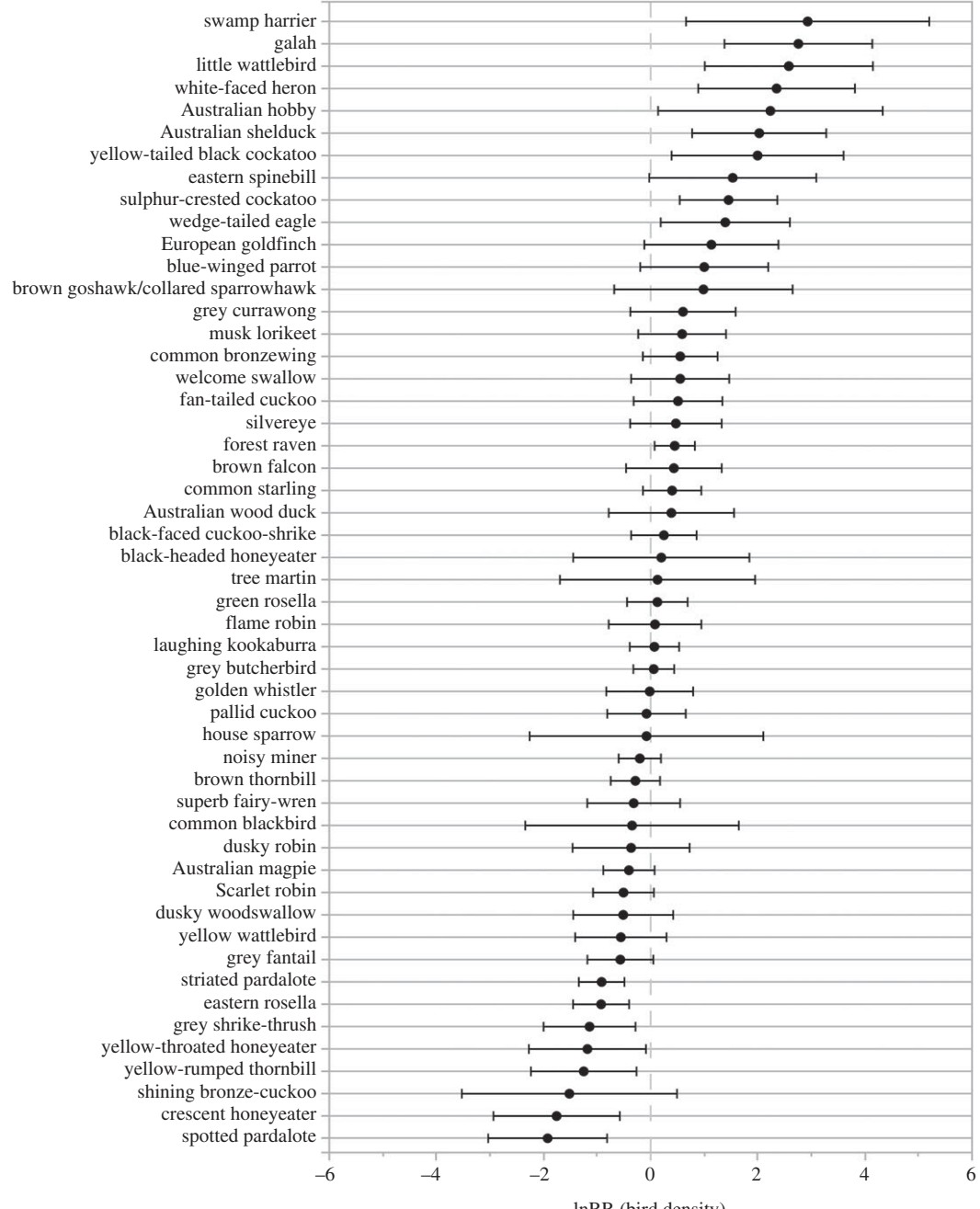

**Figure 5.** Changes in the mean density of bird species in remnant woodlands ($n = 33$) between the 1997 and 2017 survey periods. Positive values of the log-response ratio (lnRR) indicate greater abundance in 2017 while negative values indicate greater abundance in the 1997 survey period. Differences are significant when 95% CI error bars do not overlap zero. Only those species that were recorded at more than one site and for which more than five individuals were observed are included. The lnRR could not be calculated for species that were recorded in only one of the two survey periods (e.g. satin flycatcher, little and long-billed corellas) and so these birds were also excluded.

common starling had the greatest absolute increases in density: 0.26–1.11 cockatoos ha$^{-1}$ and 1.25–1.87 starlings ha$^{-1}$. Forest ravens (*Corvus tasmanicus*) were also more abundant and, as in the 1997 survey period, this species was recorded at every survey site.

Of the 10 species with the lowest response ratios (i.e. less abundant in the recent survey), two are endemic to Tasmania (yellow-throated honeyeater *Lichenostomus flavicollis* and yellow wattlebird *Anthochaera paradoxa*) and eight are arboreal foragers. The exceptions were eastern rosellas and the yellow-rumped thornbill (*Acanthiza chrysorrhoa*), both of which are classed as ground-foraging species. At historical sites, three species were recorded on transects in the 1997 survey period but were absent in the present survey: masked lapwing (*Vanellus miles*), peregrine falcon and chestnut teal (*Anas castanea*).

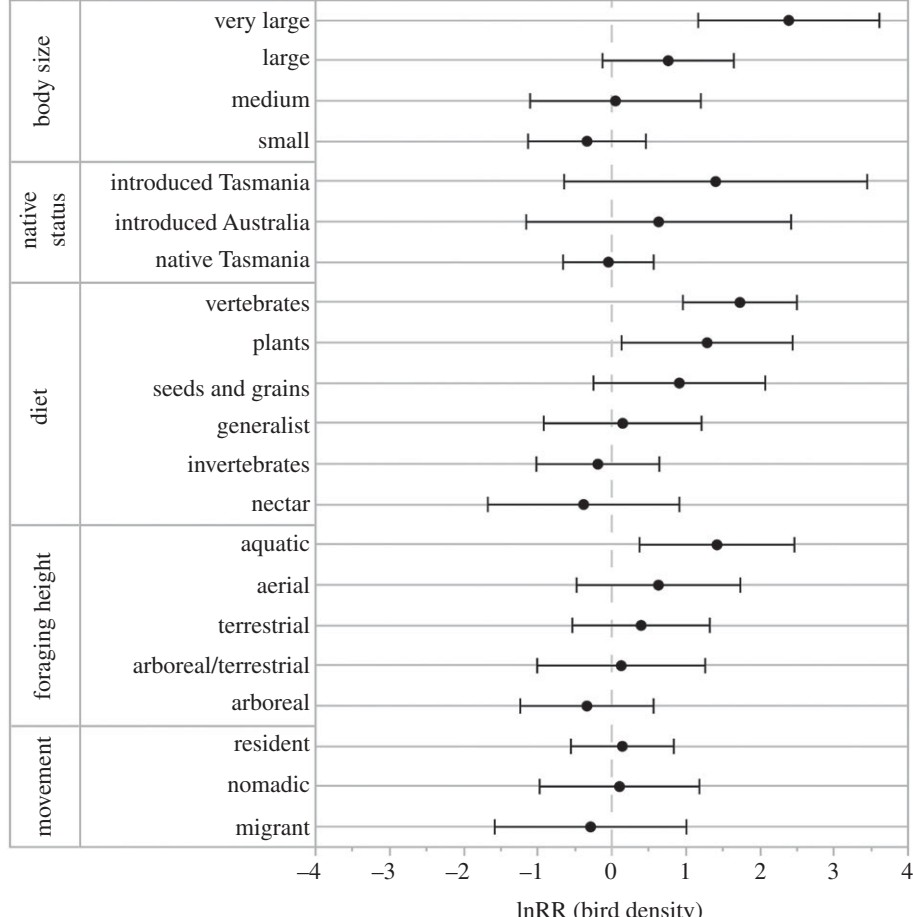

**Figure 6.** Changes in the density of all bird species that shared traits. Positive values of the log-response ratio (lnRR) indicate greater abundance in the 2017 survey period. Negative values indicate greater abundance in the 1997 survey period. Error bars are 95% confidence intervals.

Conversely, four species were recorded in the current survey but were previously undetected: satin flycatcher (*Myiagra cyanoleuca*), little corella, long-billed corella and the Pacific black duck (*Anas superciliosa*).

## 4. Discussion

The Midlands region of Tasmania is a microcosm of agricultural landscapes elsewhere in Australia and the world, with many of the same threats to avifauna. The well-defined geographical boundaries of the Midlands Biodiversity Hotspot provide an ideal opportunity for studying the impacts of land-use change on birds and other wildlife. We first examined how environmental characteristics influence current patterns of bird species presence and community composition, before considering how this might have changed as a result of agricultural intensification. The composition of bird communities in remnant woodlands of the Midlands was strongly influenced by the amount of surrounding woody vegetation, landcover diversity and the elevation of survey sites, but elements of structural complexity such as foliage projective cover (FPC) and leaf litter cover were also important. As in other studies of Australian avifauna, the presence of aggressive noisy miners had strong negative effects on the abundance and richness of small birds [39,53], highlighting what will be a key management challenge for local restoration efforts. Agricultural intensification in the Midlands over the past 20 years appears to have favoured large-bodied birds; populations of some small and medium-sized species, particularly arboreal foragers, could be in decline.

### 4.1. Bird community composition and habitat variables

The greatest gains in avian biodiversity are to be made by restoring habitat in landscapes with low levels of native vegetation cover [54]. We found that increasing levels of wooded cover could have significant

benefits for native birds in our study region. Survey sites surrounded by more woodland supported a more diverse bird community with a higher abundance of nectarivores and generalist foragers and fewer exotic species. Four of the seven species with significantly lower abundances in the 2017 survey period (spotted pardalote *Pardalotus punctatus*, crescent honeyeater *Phylidonyris pyrrhopterus*, yellow-throated honeyeater, grey shrike-thrush *Colluricincla harmonica*) were also positively associated with woody vegetation cover. Thus, restoring tree cover might especially benefit these species, which are potential targets for ecological restoration. Species–area relationships have been well described in birds [55–59]. Large patches of woodland and landscapes with high vegetation cover provide more resources, such as space, food and safe nesting sites, as well as better access to these resources, and can, therefore, support larger and more diverse bird populations [60–62]. Increasing vegetation cover might also improve the ability of birds to colonize remnant vegetation, mitigate negative edge effects and encourage settlement by greater numbers of migratory individuals [63–65]. Nonetheless, small patches of remnant habitat still contribute to avian biodiversity in fragmented landscapes [59]. This was true for small woodlands and planting sites in the Midlands but only when these habitats were free from high densities of noisy miners.

There was a clear division between bird communities in noisy miner-dominated woodlands, which were mainly comprised of large species characteristic of open-farmland environments (e.g. Australian magpie, laughing kookaburra), and woodlands without miners where smaller species could persist. This pattern is consistent with the earlier survey of the Midlands [25] and is common to fragmented habitat throughout Australia wherever aggressive *Manorina* honeyeaters are present [39,66,67]. Our study adds that temporal changes in miner density could explain shifts between these two types of bird community. Two woodlands demonstrated large changes in species richness; one increased by 18 bird species and the other declined in richness by the same number. Miner densities at these sites changed from 1.75 to 0.56 miners ha$^{-1}$ and 0.56 to 1.86 miners ha$^{-1}$, respectively. Thomson *et al.* [39] had previously estimated 0.56 miners ha$^{-1}$ as the threshold above which miners in the Northern Midlands bioregion were expected to negatively affect the occupancy of small birds. Thus, such dramatic changes in community composition are consistent with the impact threshold proposed by Thomson *et al.* [39]; this represents an effective target for the maximum density of noisy miners that could be tolerated by other birds in restored habitats. Methods of controlling miner populations are currently the focus of much scientific inquiry ([68], but see Recommendations for restoration).

FPC and leaf litter cover also had significant effects on community composition. Arboreal foragers and species more typical of wet forests (e.g. satin flycatcher & Tasmanian scrubwren *Sericornis humilis*) were most abundant at sites with high FPC while granivorous species and aerial foragers showed the opposite response. High FPC and leaf litter cover might reflect greater site productivity (FPC was positively correlated with mean annual rainfall) and a more abundant and diverse prey-base for insectivorous birds [69–71]. Certainly, all three species of robin included in our analysis showed positive associations with leaf litter cover. Sites with high FPC might also provide better opportunities for nesting sites and greater protection to birds from introduced predators like feral cats [72]. Birds that were negatively associated with leaf litter cover tended to be more common in degraded woodlands where pasture grasses were the dominant groundcover (e.g. forest raven and common starling).

Elevation had a strong influence on community composition but only one species, the grey fantail, was found to contribute significantly to this effect. The apparent negative relationship between grey fantails and elevation could be the result of seasonal migratory behaviour in this species, which is poorly understood in Tasmania [73]. Some birds, such as the crescent honeyeater, are known to migrate altitudinally in response to changes in food availability but individual trends for these elevational migrants could have been masked in our analysis because we combined survey data from different seasons. Given that our study sites were mainly restricted to the rift valley of the Midlands, our surveys were not designed to reveal the full effects of elevation on Tasmanian bird assemblages. Nonetheless, our models showed that endemics like the black currawong, black-headed honeyeater and yellow-throated honeyeater were more common at higher altitudes.

## 4.2. Traits and changes in abundance

The bird community of Tasmania is distinct from those of mainland Australia but is also relatively depauperate [25]. This could result from a dispersal filter to the island or Tasmania's climatic suitability. Of the terrestrial bird species that settled in Tasmania, and when compared to those of mainland Australia, few have been recognized as of 'conservation concern', 'decliners' or as sensitive to the area of woodland remnants [15,58,71]. For example, of the 26 species of declining woodland

bird analysed by Watson [71], only the dusky woodswallow (*Artamus cyanopterus*) and swift parrot (*Lathamus discolor*) are present in Tasmania. The paucity of research on woodland birds in Tasmania could mean that species of conservation concern are yet to be identified or might otherwise indicate that local species are more resilient to the impacts of land-use change and are not at overall risk.

Of the 51 species that we examined, seven showed significantly lower current densities than in the 1997 survey period, but without data collected from the intervening years, we cannot be certain if these differences truly reflect population declines. This is particularly so for striated pardalotes (*Pardalotus striatus*) and crescent honeyeaters, both of which have strong migratory patterns that could be influenced by yearly variations in climate. Many of those species with lower abundances share a small to medium body size and are arboreal foragers. Birds with these life-history traits may be particularly sensitive to continued habitat loss and the degradation of structural complexity in remnant woodlands, for example by livestock grazing or domestic firewood collection. Consistent with our findings, the spotted pardalote, striated pardalote, grey shrike-thrush and yellow-rumped thornbill have been identified as in regional decline elsewhere in Australia, although the reasons for their decline remain unclear [15,59,74]. Unlike Lindenmayer *et al.* [15] who found declines in numbers of large-bodied birds, we found an increase in larger species.

The diversification of farms in the Midlands from sheep grazing to cropping has favoured large-bodied granivorous birds, namely *Cacatuids*, by providing these species with a rapidly expanding food source. The proliferation of farm dams and conversion of pastures has also benefitted another large but mainly herbivorous species, the Australian shelduck. Incidental to our bird surveys, we recorded flocks of 189 sulphur-crested cockatoos and 291 shelducks foraging in pastures adjacent to our field sites. Corellas were absent in the 1997 survey period but were frequently recorded in the present study [75]. The first formal record of little corellas in Tasmania was in 1983 on a farm in the Northern Midlands that is now the focus of landscape restoration [76]. We expect that corellas will continue to increase their abundance and distribution in much the same way as the laughing kookaburra following its introduction to the Midlands in 1906 [44].

Populations of raptors and other large carnivorous species like the forest raven may have also increased. This could be because of a greater availability of animal carcasses following the functional loss of the native apex-predator, the Tasmanian devil [77], higher stocking rates of lambs, reduced persecution and use of poisons by landowners, or perhaps even habitat fragmentation itself. Aerial foragers, which were mostly raptors, showed a preference for landcover diversity and low woodland cover near to survey sites, possibly reflecting their hunting strategies.

We found a high proportional increase in species that are typically associated with wetlands. Record rainfall meant that in 2016 many of the ephemeral lagoons near to our survey sites were filled and some paddocks were flooded [30]. This attracted dispersive species of duck, white-faced herons and other migratory birds like the swamp harrier, such that they were more frequently recorded within neighbouring woodlands. By contrast, the historical survey period was undertaken at the beginning of Australia's Millennium Drought. Wetland birds could also have benefitted from the widespread creation of farm dams to support crop irrigation.

## 4.3. Recommendations for restoration

The invasion of restored habitat by noisy miners poses significant risk to the avian biodiversity objectives of landscape restoration in the Midlands. The habitat preferences of noisy miners are well established: eucalypt-dominated woodlands of high productivity [78], a high perimeter-area ratio [26,79] and low structural complexity [26,39,80]. Therefore, to abate the threat of miners, restoration practitioners should maximize levels of shrub cover [80], establish mixed stands of eucalypt and non-eucalypt tree species (e.g. *Bursaria* spp., [81]) and plant in blocks rather than corridors whenever that is feasible [79,82].

We encourage restoration managers to trial novel planting configurations and test for the most 'miner-resilient' formations. This could include plantings that are bordered by tree species, native or exotic, that have a very dense foliage [83]. Alternatively, eucalypt plantings could have higher than natural stem densities at the edge but retain an open structure within their interior. Some landowners in the Midlands regularly clear vegetation along fence lines to avoid damage by tree falls. The associated soil disturbance has led to dense resprouting of *Acacia* trees [84] around the perimeter of woodland remnants and may function to prevent colonization by noisy miners. Small islands of dense vegetation could also be dispersed within miner-dominated remnants to facilitate the movement of small birds through revegetation corridors, analogous to paddock trees as stepping stones for birds across the agricultural matrix [85].

Miner culls have also been proposed as a management action to encourage the return of small birds to remnant habitat [26,86]. Some authors have reported immediate benefits of culling miners [87,88], but more recent studies have found no short-term effects on bird species richness and abundance [66,68]. The ecological benefit of culls probably depends on the initial density of miner birds in degraded habitat [68,89]. Further research is needed to test whether culls are more successful in remnant woodlands that are contiguous with revegetation sites or when there is a combined habitat restoration effort [88].

In one small woodland remnant, the establishment of a neighbouring eucalypt plantation led to a doubling of native bird species richness. This suggests that even when revegetation is not designed with the purpose of improving biodiversity, increasing levels of woody vegetation cover can still benefit local avifauna. It is unclear, however, whether this increase in species richness was due to the provision of more habitat, improved connectivity with nearby remnants or the associated decline in noisy miner abundance. Forestry plantations could be incorporated into the design of landscape restoration. Law *et al*. [90] found that as eucalypt plantations in a farmland mosaic of northern New South Wales matured, abundance of noisy miners declined. Commercial eucalypt plantations enhance the biodiversity value of the matrix between remnants [91] and can provide suitable foraging habitat for some Australian birds [90,92–94].

Even though many of our small woodland sites were degraded and had become dominated by noisy miners, their conservation value should not be ignored. Huth & Possingham [91] showed that there is greater benefit to avian biodiversity in restoring structural attributes of small and degraded woodlands than there is in increasing their size. Moreover, connecting small remnants without first restoring structural complexity might only lead to the creation of more habitat for miners [86]. Small remnants could be improved by reducing grazing pressure from livestock or perhaps through ecological burns such that leaf litter cover and FPC are restored. Gorse shrubs are present in many woodland remnants in the Midlands and are often the only feature that provides vertical structural complexity in this habitat. As has been suggested for *Tamarix*, a genus of invasive weeds in North America, gorse should be replaced gradually by native shrubs that offer comparable vegetation cover, rather than being immediately cleared [95]. Gorse shrubs within remnants provide small birds with safe nesting sites as well as protection from noisy miners and other predators. We also recorded a range of species using gorse-invaded pastures to travel between woodland remnants, highlighting the weed's value in softening the agricultural matrix.

Large increases in the abundance of hollow-nesting birds, including sulphur-crested cockatoos, common starlings, galahs and corellas, has probably increased competition for what was an already limited resource in the Midlands. Competition for breeding hollows is thought to contribute to declining numbers of eastern rosellas in Tasmania [96] and could increasingly threaten other species like the dusky woodswallow. Tree hollows suitable for breeding birds can take more than 100 years to form in *Eucalyptus* [97] and so revegetation sites will not be able to relieve such demand in the immediate future. Artificial nest-boxes could be used to supplement natural tree hollows within remnant woodlands but are unlikely to attract hollow-breeding birds to young stands of vegetation [98].

## 5. Conclusion

Agricultural intensification in the Tasmanian Midlands appears to have benefitted some bird species. These tended to be larger birds with either a granivorous or carnivorous diet. By contrast, arboreal foragers and nectarivores that prefer areas of high woodland cover and rely on more complex vegetation structure could be in decline. A continued long-term monitoring effort is necessary to confirm changes in bird abundance because our analysis was limited to comparisons between just two survey periods. This is especially urgent considering the emerging trend of population decline in common birds around the world [5,15,99], the rapidly changing environment in the Midlands, and the poor representation of Tasmanian species in Australian bird data. In a landscape like the Midlands, which has very little remaining native vegetation, simply increasing levels of wooded cover will have significant benefits for local bird populations [100]. To maximize the conservation value of restoration efforts, however, competition between birds and the specific habitat preferences of individual species must also be addressed.

Ethics. All birds were surveyed under the University of Tasmania Animal Ethics approval no. A0014880. All research was conducted with the permission of landowners and under scientific permit from the Department of Primary Industries, Parks, Water and Environment, authority no. FA15132.

Data accessibility. Datasets used in this study are available through figshare at https://doi.org/10.6084/m9.figshare.9192485. Bird survey data has been uploaded to the Natural Values Atlas of Tasmania.

Authors' contribution. G.C.B. collected the field data, helped conceive the study, performed the analysis and wrote the paper; R.G. and R.H. helped collect the vegetation data and discussed the results and statistical analyses; M.A.M. collected the historical field data and participated in the design of the study; C.N.J. and M.E.J. helped to conceive and design the study, discussed the results and statistical analyses and helped draft the manuscript. All authors revised the manuscript and gave final approval for publication.

Competing interests. We declare we have no competing interests.

Funding. M.E.J. is the recipient of an Australian Research Council Future Fellowship (grant no. FT100100031) and C.N.J. was supported by an ARC Australian Professorial Fellowship (grant no. DP110103069). The Midlands Restoration Project is supported by an ARC Linkage (grant no. LP130100949) to M.E.J. and C.N.J.

Acknowledgements. We thank the many landowners who gave permission for surveys to be conducted on their properties. We are especially appreciative for the kind hospitality of Rae and Lindsay Young. We thank our industry partners: Neil Davidson, Sebastian Burgess and Tanya Bailey (formerly Greening Australia) and collaborators: Matt Appleby (Bush Heritage Australia), Daniel Sprod (formerly Tasmanian Land Conservancy), Louise Gilfedder and Oberon Carter (Department of Primary Industry, Parks, Water and Environment) for their ongoing logistical and intellectual support. Finally, we are grateful to Joanne Potts for providing statistical expertise and to anonymous referees for their feedback on earlier versions of the manuscript.

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
