## [Reviewer comments · Royal Society Open Science]

Review History

RSOS-191405.R0 (Original submission)

Review form: Reviewer 1

Is the manuscript scientifically sound in its present form?

No

Are the interpretations and conclusions justified by the results?

No

Is the language acceptable?

Yes

Do you have any ethical concerns with this paper?

No

Have you any concerns about statistical analyses in this paper?

Yes

Recommendation?

Reject

Comments to the Author(s)

This manuscript, by Bain and collaborators, summarises a retrospective series of comparisons of bird occurrence in the midlands of Tasmania, as part of ongoing restoration efforts aimed to improve wildlife habitats in the region. The basis of the work is two sets of bird surveys collected at thirty three sites in 1996/1997 and 2015/2017, with another 39 sites added for the recent round of surveys. This manuscript summarized a tremendous amount of work, both in the field conducting bird surveys and vegetation data, compiling site attributes in a GIS, and analysing determinants of bird occurrence.

I have four sets of substantive comments, relating to overall study design, bird surveying methods, analysis and inference as well as various minor quibbles.

1. Study design. Using effectively two data points (occurrence patterns of birds in time period 1 and 2, twenty years apart) makes it very hard to identify trends, especially for an assemblage of animals characterized for their high vagility and often broad habitat tolerances. Rather than using this temporal dimension to frame their analyses, I would suggest running the same sets of analyses for the two datasets and looking instead at how determinants of occurrence / richness have changed, and exploring how those differences might relate to the dramatic land use changes that have occurred in the intervening period.

2. While I appreciate the method used to estimate bird occurrence was effectively decided by MacDonald and Kirkpatrick for the 1996/97 survey period, many of the concerns I have with the data relate to how it has been handled, rather than initially collected. Variable effort was used which is not necessarily problematic. Rather, by adjusting their data based on proportion of the site surveyed, a systematic error forcing a specie-area relationship might be the real problem. This can be avoided by using estimated richness rather than observed richness, following the methods of Colwell in his freely available EstimateS software. They can then look at how much the predicted richness diverges from observed richness and, if there are consistent effects of sampling effort, they can be quantified and more precisely dealt with. As for density, I have even more concern with the response variable used, and would recommend the authors follow the work of Mac Nally and others and uses incidence or reporting rate instead. That is, the proportion of surveys conducted in that site in which the species was detected as present. It is coarse, but far less prone to the many confounding effects of variable detectability, (between species, sites and seasons).

3. The analyses used were complex and, although well explained, the authors side-stepped what I took away as one of their main findings – the avifauna changed very little between sampling periods, and most of the factors measured had little explanatory power, either at the individual species scale or for variously-defined groups. There was also a very large number of potential relationships to explore--figure 4 alone is a distillation of approximately 600 univariate relationships. The determinants of species richness very murky, with more than half the variation unexplained. Yes, this is a negative result, but if that result remains using predicted richness where comparability across sites is assured, then the authors should stand by it, and discuss this lack of signal more explicitly. The fact that these site-based factors had so little explanatory power suggests that landscape scale factors might be more important, that inter-specific differences and the masking effects of simultaneous positive and negative relationships might conceal any assemblage-wide trajectories.

4. Related to point 3, I would suggest the authors be a little more circumspect about their inferences. In figure 5, temporal changes for 33 species are depicted of which more than half have confidence intervals that overlap with zero. To me, this suggests that either most species aren't changing or the approach used to monitor them across this period is unable to discern any change. Likewise, in figure 6: just four of the 21 trait-defined groups exhibited significant changes in density: waterbirds, raptors, large birds and plant eaters. Rather than any trait-based influence, these four groups likely reflect the clear increases noted in swamp harriers and shelducks.

I would recommend the authors consider subdividing this large manuscript into three more narrowly-defined contributions. The noisy miner story might well be worth carving off and exploring on its own. [Oh, and please don't use the term "reverse keystone". Keystones are by definition rare in their communities, in terms of numbers of individuals / proportion of biomass - the term "despotic" is far more appropriate]. The determinants of diversity in two different time periods would also stand alone, leaving perhaps a third manuscript considering what you've found overall, what appears to be important and not important in structuring these bird assemblages in a highly modified system, how you might expect birds to respond to planned restoration, and some guiding predictions to revisit in another twenty years. This is all up to you but by lumping all of this into one paper, not only does it become unwieldy, but you simply don't have the space to consider your findings. There's plenty of interesting comments made in passing about particular sites, land-use practices, positive and negative responses to shifting land use. Reworking the paper into more clearly defined narratives will give you the structure and space to explore these in more depth, and make better use of all the hard work that went into compiling all of this information.

Review form: Reviewer 2

Is the manuscript scientifically sound in its present form?

Yes

Are the interpretations and conclusions justified by the results?

Yes

Is the language acceptable?

Yes

Do you have any ethical concerns with this paper?

No

Have you any concerns about statistical analyses in this paper?

No

Recommendation?

Accept with minor revision (please list in comments)

Comments to the Author(s)

Review: Long - Long-term change in bird communities of an agricultural landscape: declines in arboreal foragers, increases in large species.

Bain et al use bird surveys (past and present) to compare species and their traits with habitat characteristics in the Tasmanian midlands. Over all I think the paper is well written, the methods appropriate and well executed and the results interesting and well interpreted. I particularly like the use of the fourth corner approach (and the resulting figures) to addressing the question of how to link species occurrence data with complex habitat variation. My comments on are relatively minor, but I would suggest that the authors look out for over-long sentences in their revision, and attempt to clarify some parts of the methods as suggested below. The manuscript could be trimmed in some places to save words.

Methods - generally fine, but more detail is needed about the method used in the original surveys - given the time that has passed since the first surveys, it is important to understand the possible factors that could confound the results between the past and now. E.g. was observer error quantified and if not, then this assumption should be spelled out more (the first mention I

see is at L428 which is a bit late). I see GB did all the field work in the contemporary surveys- who did them in the past? Is it possible to quantify detection error between the surveyors?

L157: presumably the orientation and location of transects was different due to change in vegetation cover (or not?) between survey periods? If yes, say so.

L158: if I understand correctly, small patches received 1 * 100m survey transect and large ones received 4 * 200m transects back-to-back over 800m, plus an addition 20min/2ha? was the orientation random in large woodlands? Why you used 2 survey approaches (transects and then 2ha searches) - did you pool the data for analysis or were they used for different parts of the analysis? It's not clear to me how these different methods are used in analysis

L198: "as a proxy for land use intensity."

L204: specify here where you sourced miner density estimates - is this derived from your own models?

L265: I understand this limitation of mvabund but think it would be helpful for readers if you provided some explanation of what this might mean for interpreting your potential results, especially since the large forest patches had quadruple the effort at transects twice as long as those in small patches

L270: Clarify species richness - is that just the number of species per survey? Or number per site over the survey periods?

Fig 4. It would help to have the environmental variable names on both the top and bottom for readability, but not critical

L413-17: long and slightly clunky sentence. Reword.

L440 and L671: remove 'productive' - plantations are referred to in other ways too elsewhere in the paper - be consistent

L465: "Granivores and raptors appeared..."

L468: re 'common in urban environments' either needs a citation or delete (relevance to the section seems tangential anyway)

L611: flock sizes are results

Review form: Reviewer 3 (Richard H. Loyn)

Is the manuscript scientifically sound in its present form?

Yes

Are the interpretations and conclusions justified by the results?

Yes

Is the language acceptable?

Yes

Do you have any ethical concerns with this paper?

No

Have you any concerns about statistical analyses in this paper?

No

Recommendation?

Accept with minor revision (please list in comments)

Comments to the Author(s)

A very good paper. See attached file for some comments (Appendix A).

Decision letter (RSOS-191405.R0)

18-Sep-2019

Dear Mr Bain,

Manuscript ID RSOS-191405 entitled "Long-term change in bird communities of an agricultural landscape: declines in arboreal foragers, increases in large species." which you submitted to Royal Society Open Science, has been reviewed. The comments from reviewers are included at the bottom of this letter.

In view of the criticisms of the reviewers, the manuscript has been rejected in its current form. However, a new manuscript may be submitted which takes into consideration these comments.

Please note that resubmitting your manuscript does not guarantee eventual acceptance, and that your resubmission will be subject to peer review before a decision is made.

Your resubmitted manuscript should be submitted by 17-Mar-2020. If you are unable to submit by this date please contact the Editorial Office.

Kind regards,
Lianne Parkhouse
Royal Society Open Science
openscience@royalsociety.org

on behalf of the Associate Editor and Professor Kevin Padian (Subject Editor)
openscience@royalsociety.org

Editor Comments to the Author:

Thanks for your very interesting submission. The reviewers have a variety of comments that reflect different concerns, although they are generally positive. However, it would seem important to show specifically what is a new advance of your research, and also to address the question whether you might want to make this two papers (not required, but please address the question). Please see our guidelines for resubmissions as they may help you in this respect. Best success in your revisions.

Associate Editor Comments to Author:

The reviewers have provided the Editors - and hopefully yourselves - with some food for thought. Each reviewer sees value in your work, and each recommend modifications of varying degrees to improve the current paper, the most critical of the reviewers recommends a more radical step: splitting the manuscript into two parts to tell the two distinct but related stories they have identified in your submission. It is on this basis that a reject/resubmit decision has been issued (not a reflection of the quality of the work per se).

We'd like to give you the choice of either revising the current manuscript to address as far as possible the concerns raised during this round of review and so get the paper across the line largely in its current format (we'll ask the existing reviewers to comment on the changes made).

Or you may take the opportunity to divide your work as suggested by the reviewer and use the resubmission as the basis for one of the two papers, and then preparing a second manuscript for submission and consideration alongside the resubmission (again, we'll ask the current reviewers for their thoughts).

Whichever approach you take, we'd be grateful if you could contact the editorial office to let them know, so we can take the appropriate steps post-revision.

Good luck!

Reviewers' Comments to Author:

Reviewer: 1

This manuscript, by Bain and collaborators, summarises a retrospective series of comparisons of bird occurrence in the midlands of Tasmania, as part of ongoing restoration efforts aimed to improve wildlife habitats in the region. The basis of the work is two sets of bird surveys collected at thirty three sites in 1996/1997 and 2015/2017, with another 39 sites added for the recent round of surveys. This manuscript summarized a tremendous amount of work, both in the field conducting bird surveys and vegetation data, compiling site attributes in a GIS, and analysing determinants of bird occurrence.

I have four sets of substantive comments, relating to overall study design, bird surveying methods, analysis and inference as well as various minor quibbles.

1. Study design. Using effectively two data points (occurrence patterns of birds in time period 1 and 2, twenty years apart) makes it very hard to identify trends, especially for an assemblage of animals characterized for their high vagility and often broad habitat tolerances. Rather than using this temporal dimension to frame their analyses, I would suggest running the same sets of analyses for the two datasets and looking instead at how determinants of occurrence / richness have changed, and exploring how those differences might relate to the dramatic land use changes that have occurred in the intervening period.

2. While I appreciate the method used to estimate bird occurrence was effectively decided by MacDonald and Kirkpatrick for the 1996/97 survey period, many of the concerns I have with the data relate to how it has been handled, rather than initially collected. Variable effort was used which is not necessarily problematic. Rather, by adjusting their data based on proportion of the site surveyed, a systematic error forcing a specie-area relationship might be the real problem. This can be avoided by using estimated richness rather than observed richness, following the methods of Colwell in his freely available EstimateS software. They can then look at how much the predicted richness diverges from observed richness and, if there are consistent effects of sampling effort, they can be quantified and more precisely dealt with. As for density, I have even more concern with the response variable used, and would recommend the authors follow the work of Mac Nally and others and uses incidence or reporting rate instead. That is, the proportion of surveys conducted in that site in which the species was detected as present. It is coarse, but far less prone to the many confounding effects of variable detectability, (between species, sites and seasons).

3. The analyses used were complex and, although well explained, the authors side-stepped what I took away as one of their main findings – the avifauna changed very little between sampling periods, and most of the factors measured had little explanatory power, either at the individual species scale or for variously-defined groups. There was also a very large number of potential relationships to explore--figure 4 alone is a distillation of approximately 600 univariate

relationships. The determinants of species richness very murky, with more than half the variation unexplained. Yes, this is a negative result, but if that result remains using predicted richness where comparability across sites is assured, then the authors should stand by it, and discuss this lack of signal more explicitly. The fact that these site-based factors had so little explanatory power suggests that landscape scale factors might be more important, that inter-specific differences and the masking effects of simultaneous positive and negative relationships might conceal any assemblage-wide trajectories.

4. Related to point 3, I would suggest the authors be a little more circumspect about their inferences. In figure 5, temporal changes for 33 species are depicted of which more than half have confidence intervals that overlap with zero. To me, this suggests that either most species aren't changing or the approach used to monitor them across this period is unable to discern any change. Likewise, in figure 6: just four of the 21 trait-defined groups exhibited significant changes in density: waterbirds, raptors, large birds and plant eaters. Rather than any trait-based influence, these four groups likely reflect the clear increases noted in swamp harriers and shelducks.

I would recommend the authors consider subdividing this large manuscript into three more narrowly-defined contributions. The noisy miner story might well be worth carving off and exploring on its own. [Oh, and please don't use the term "reverse keystone". Keystones are by definition rare in their communities, in terms of numbers of individuals / proportion of biomass- the term "despotic" is far more appropriate]. The determinants of diversity in two different time periods would also stand alone, leaving perhaps a third manuscript considering what you've found overall, what appears to be important and not important in structuring these bird assemblages in a highly modified system, how you might expect birds to respond to planned restoration, and some guiding predictions to revisit in another twenty years. This is all up to you but by lumping all of this into one paper, not only does it become unwieldy, but you simply don't have the space to consider your findings. There's plenty of interesting comments made in passing about particular sites, land-use practices, positive and negative responses to shifting land use. Reworking the paper into more clearly defined narratives will give you the structure and space to explore these in more depth, and make better use of all the hard work that went into compiling all of this information.

Reviewer: 2

Review: Long - Long-term change in bird communities of an agricultural landscape: declines in arboreal foragers, increases in large species.

Bain et al use bird surveys (past and present) to compare species and their traits with habitat characteristics in the Tasmanian midlands. Over all I think the paper is well written, the methods appropriate and well executed and the results interesting and well interpreted. I particularly like the use of the fourth corner approach (and the resulting figures) to addressing the question of how to link species occurrence data with complex habitat variation. My comments on are relatively minor, but I would suggest that the authors look out for over-long sentences in their revision, and attempt to clarify some parts of the methods as suggested below. The manuscript could be trimmed in some places to save words.

Methods – generally fine, but more detail is needed about the method used in the original surveys – given the time that has passed since the first surveys, it is important to understand the possible factors that could confound the results between the past and now. E.g. was observer error quantified and if not, then this assumption should be spelled out more (the first mention I see is at L428 which is a bit late). I see GB did all the field work in the contemporary surveys- who did them in the past? Is it possible to quantify detection error between the surveyors?

L157: presumably the orientation and location of transects was different due to change in vegetation cover (or not?) between survey periods? If yes, say so.

L158: if I understand correctly, small patches received 1 * 100m survey transect and large ones received 4 * 200m transects back-to-back over 800m, plus an addition 20min/2ha? was the

orientation random in large woodlands? Why you used 2 survey approaches (transects and then 2ha searches) – did you pool the data for analysis or were they used for different parts of the analysis? It's not clear to me how these different methods are used in analysis

L198: "as a proxy for land use intensity."

L204: specify here where you sourced miner density estimates – is this derived from your own models?

L265: I understand this limitation of mvabund but think it would be helpful for readers if you provided some explanation of what this might mean for interpreting your potential results, especially since the large forest patches had quadruple the effort at transects twice as long as those in small patches

L270: Clarify species richness – is that just the number of species per survey? Or number per site over the survey periods?

Fig 4. It would help to have the environmental variable names on both the top and bottom for readability, but not critical

L413-17: long and slightly clunky sentence. Reword.

L440 and L671: remove 'productive' – plantations are referred to in other ways too elsewhere in the paper - be consistent

L465: "Granivores and raptors appeared..."

L468: re 'common in urban environments' either needs a citation or delete (relevance to the section seems tangential anyway)

L611: flock sizes are results

Reviewer: 3

A very good paper. See attached file for some comments.

Author's Response to Decision Letter for (RSOS-191405.R0)

See Appendices B & C.

RSOS-200076.R0

Review form: Reviewer 1

Is the manuscript scientifically sound in its present form?

Yes

Are the interpretations and conclusions justified by the results?

No

Is the language acceptable?

Yes

Do you have any ethical concerns with this paper?

No

Have you any concerns about statistical analyses in this paper?

No

Recommendation?

Accepted with minor revision (please list in comments)

Comments to the Author(s)

...

Decision letter (RSOS-200076.R0)

03-Feb-2020

Dear Mr Bain

On behalf of the Editor, I am pleased to inform you that your Manuscript RSOS-200076 entitled "Changing bird communities of an agricultural landscape: declines in arboreal foragers, increases in large species." has been accepted for publication in Royal Society Open Science subject to minor revision in accordance with the referee suggestions. Please find the referees' comments at the end of this email.

The reviewers and Subject Editor have recommended publication, but also suggest some minor revisions to your manuscript. Therefore, I invite you to respond to the comments and revise your manuscript.

- Ethics statement

- Data accessibility

If you wish to submit your supporting data or code to Dryad (<http://datadryad.org/>), or modify your current submission to dryad, please use the following link:
<http://datadryad.org/submit?journalID=RSOS&manu=RSOS-200076>

- Competing interests

- Authors' contributions

- Acknowledgements

- Funding statement

Because the schedule for publication is very tight, it is a condition of publication that you submit the revised version of your manuscript before 12-Feb-2020. Please note that the revision deadline will expire at 00.00am on this date. If you do not think you will be able to meet this date please let me know immediately.

on behalf of Kevin Padian (Subject Editor)
openscience@royalsociety.org

Author's Response to Decision Letter for (RSOS-200076.R0)

See Appendix D.

Decision letter (RSOS-200076.R1)

14-Feb-2020

Dear Mr Bain,

It is a pleasure to accept your manuscript entitled "Changing bird communities of an agricultural landscape: declines in arboreal foragers, increases in large species." in its current form for publication in Royal Society Open Science. The comments of the reviewer(s) who reviewed your manuscript are included at the foot of this letter.

on behalf of Mr Andrew Dunn (Associate Editor) and Kevin Padian (Subject Editor)
openscience@royalsociety.org

Associate Editor Comments to Author (Mr Andrew Dunn):
Associate Editor
Comments to the Author:
(There are no comments.)

Reviewer comments to Author:

Appendix A

Paper on 20-year changes in woodland birds in Tasmanian Midlands

By Glen Bain et al for Roy Soc Open Science

This is an excellent paper on a subject of global interest. It is well written, and the literature view is very thorough for recent studies. The paper can be published with very little change. However, the following points could add some value:

1. The term “response ratio” could imply a response to an identified process such as agricultural intensification, whereas in fact it just refers to a change over time between the two survey periods (with agricultural intensification as one of several possible causal factors). Would “percent change” be a less misleading term?
2. The trait-based approach is clearly useful, and quite similar to the guild approach used in various similar studies. The latter approach would enable conclusions to be made about changes in the abundance of whole guilds, e.g. nectarivores, canopy-foraging insectivores or small birds and large birds, rather than just saying that several species with those traits increased or decreased. It seems that calculations were made along these lines (Fig. 6) and perhaps more could be made of them in the text? I found only one reference to Fig. 6, and that just related to large herbivores such as ducks (line 465).
3. The basic results (as shown in Figure 2) are similar to those from a study of fragmented forests in south-eastern Victoria (Loyn 1985, 1987), which showed the role of noisy miners, and gave estimates of likely declines in numbers of species over time. The study was repeated 22 years later, and the further declines were found to be less than predicted at a set of sites where eucalypt plantations had been established nearby (MacHunter et al 2006). It may be worth comparing these results.

Some specific comments (re line numbers):

Title. Not sure that 20 years is long-term. Perhaps “Changes over 20 years....”?

16. This invites questions about whole guilds (point 2 above).

27 & 28. “have” not “has” (subject is plural, “levels”), one of just two typos noticed.

92-93. Is it possible to give a reference for the use of Bass Strait islands as stopover sites? Stephen Garnett et al. wrote a paper on this in the 1980s.

108. Clarify which years were Tasmania’s coldest winter (presumably 2017?) and warmest summer (2017-18?).

207. Could also refer to the early work on this subject: Dow (1977) showed that Noisy Miners exclude other birds, and Loyn (1985) showed the importance of this process in fragmented forests in rural landscapes (in Victoria).

298. Interesting that Striated Fieldwrens were recorded at all (albeit offsite). In Victoria I have also found them in rural environments (in weedy young pine plantations and woody weeds in farmland) but quite rarely (and not for many years): they remain common in or near saltmarsh and sparsely treed heath, but otherwise rare in farmland away from the coast. [This is just a comment for

interest: no changes needed unless you wish to say something about the habitat where they were found.]

281, 473 & Figure captions. see point 1 above.

300-400. see point 3 above.

351-352. The paper by Maron et al (2013) mentioned that there was weak evidence for large-bodied birds benefitting from Noisy Miners (and strong evidence for small-bodied birds being disadvantaged, as already mentioned in this paper). It's good to see the Tasmanian study providing corroborative evidence with respect to two of the large-bodied species. (I've also found evidence for that, in recent studies in the Victorian mallee.)

512. Similar effects were also found by Carla Catterall in peri-urban and rural environments in south-east Queensland (Catterall et al. 1997, 1998).

541-542. Similar shifts in bird communities have been observed in various studies. However, this particular example is interesting (and it is remarkable that the threshold fits so well with the predictions by Thompson et al.) so I think it's worth discussing, despite being based on just two sites.

586-590. This last sentence may need rethinking. It could be misread as suggesting that some of Watson's other declining species actually do inhabit Tasmania (whereas the point is there may not have been enough monitoring to detect declines in the species known to be there). And is that really right? There are plenty of records in Birddata, which have been analysed at various times for Tasmanian bird reports and two recent reports on "the state of Tasmania's birds" (Newman et al. 2017, 2019). Perhaps it's better to just say that this points to the need for more monitoring in Tasmania, in a range of environments including rural and forested landscapes, and then make a considered comment on what the current data tell us about resilience. (There have been recent suggestions that several Tasmanian non-endemic species may be declining, including Blue-winged Parrot and Flame Robin: it's pleasing to see that the current dataset suggests the reverse for Blue-winged Parrot, and little change for Flame Robin.)

614. Did you class corellas as introduced species? I've tended to think of them as natural recent colonists (as they are in southern Victoria), and this is supported by the first record of Little Corella being in 1983 (a major drought year in SE Australia). However, I notice they are listed as introduced to Tasmania in the State of Tasmanian Birds reports: could be worth checking.

635-665. A study in Victoria found that Noisy Miners rarely use young commercial or amenity eucalypt plantations (Loyn et al. 2007), though parallel work found that they were often present in mature linear planted shelterbelts.

665. A positive effect of new eucalypt plantations was also observed by MacHunter et al. (2006).

656. There were two papers by Grey et al. reporting removal experiments on different sets of sites. The second paper is Grey et al. (1998). There have also been several removal experiments with Bell Miners, with similar results.

689. the second typo, add apostrophe ("weed's").

References (if not already in text):

No need to include them all, but some could be useful for this or subsequent papers.

- Catterall, C.P., Kingston, M.B. and Park, K. 1997 Use of remnant forest habitat by birds during winter in subtropical Australia: patterns and processes. *Pacific Conservation Biology* 3:262-74
- Catterall, C.P., Kingston, M.B., Park, K. and Sewell, S. 1998 Effects of clearing lowland eucalypt forests on a regional bird assemblage. *Biological Conservation* 84:65-81
- Dow, D.D. 1977 Indiscriminate interspecific aggression leading to almost sole occupancy of space by a single species of bird. *Emu* 77:115-21.
- Grey, M.J., Clarke, M.F. and Loyn, R.H. 1998. Influence of the Noisy Miner *Manorina melanocephala* on avian diversity and abundance in remnant Grey Box woodland. *Pacific Conservation Biology* 4: 55-69.
- Loyn, R.H. 1985. Birds in fragmented forests in Gippsland, Victoria. Chapter 31 in Keast, A., Recher, H.F., Ford, H.A. and Saunders, D. (eds), pp 324-331. *Birds of the Eucalypt Forests and Woodlands: ecology, conservation and management*. Surrey-Beatty, Sydney.
- Loyn, R.H. 1987. Effects of patch area and habitat on bird abundances, species numbers and tree health in fragmented Victorian forests. Chapter 6 in Saunders, D.A., Arnold, G.W., Burbidge, A.A., and Hopkins, A.J.M. (eds), *Nature Conservation: the role of remnants of native vegetation*, pp. 65-77. Surrey Beatty & Sons, Sydney, 410 pp.
- Loyn, R.H., McNabb, E.G., Macak, P. and Noble, P. 2007. Eucalypt plantations as habitat for birds on previously cleared farmland in south-eastern Australia. *Biological Conservation* 137: 533-548. Doi:10.1016/j.biocon.2007.03.012.
- MacHunter, J., Wright, W., Loyn, R. and Rayment, P. 2006. Bird declines over 22 years in forest remnants in south-eastern Australia: evidence of faunal relaxation? *Canadian Journal of Forest Science* 36: 2756-2768.
- Newman, M., Ramshaw, N., Walter, A., Webber, W., Drake, S. and Woehler, E. 2017. The state of Tasmania's terrestrial birds 2014-15. *Birdlife Tasmania*.
- Newman, M., Ramshaw, N., Drake, S., Woehler, E., Walter, A. and Webber, W., 2019. The state of Tasmania's birds 2015-16. *Birdlife Tasmania*.

Appendix B

Manuscript ID: RSOS-191405

Long-term change in bird communities of an agricultural landscape: declines in arboreal foragers, increases in large species.

We thank the three reviewers of this manuscript for their helpful comments. Please see our detailed responses to their suggestions below.

Please note that while revising this manuscript, the paper was also reviewed by an additional two reviewers as part of the first authors (GCB) PhD thesis submission. The thesis examiners were also experts on land-use change effects on birds. In addition to addressing the comments from reviewers for *Open Science*, we have also made some changes at the suggestion of thesis examiners. These are listed at the foot of our response.

Associate Editor Comments to Author:

The reviewers have provided the Editors - and hopefully yourselves - with some food for thought. Each reviewer sees value in your work, and each recommend modifications of varying degrees to improve the current paper, the most critical of the reviewers recommends a more radical step: splitting the manuscript into two parts to tell the two distinct but related stories they have identified in your submission. It is on this basis that a reject/resubmit decision has been issued (not a reflection of the quality of the work per se).

We'd like to give you the choice of either revising the current manuscript to address as far as possible the concerns raised during this round of review and so get the paper across the line largely in its current format (we'll ask the existing reviewers to comment on the changes made).

Or you may take the opportunity to divide your work as suggested by the reviewer and use the resubmission as the basis for one of the two papers, and then preparing a second manuscript for submission and consideration alongside the resubmission (again, we'll ask the current reviewers for their thoughts).

Whichever approach you take, we'd be grateful if you could contact the editorial office to let them know, so we can take the appropriate steps post-revision.

Good luck!

Thank you for giving us the option to resubmit the manuscript as two separate works. After considerable thought, though, we have decided that the paper is stronger as one. Please see comment 5 below for our reasoning.

Reviewer: 1

This manuscript, by Bain and collaborators, summarises a retrospective series of comparisons of bird occurrence in the midlands of Tasmania, as part of ongoing restoration efforts aimed to improve wildlife habitats in the region. The basis of the work is two sets of bird surveys collected at thirty three sites in 1996/1997 and 2015/2017, with another 39 sites added for the recent round of

surveys. This manuscript summarized a tremendous amount of work, both in the field conducting bird surveys and vegetation data, compiling site attributes in a GIS, and analysing determinants of bird occurrence.

I have four sets of substantive comments, relating to overall study design, bird surveying methods, analysis and inference as well as various minor quibbles.

- 1) Study design. Using effectively two data points (occurrence patterns of birds in time period 1 and 2, twenty years apart) makes it very hard to identify trends, especially for an assemblage of animals characterized for their high vagility and often broad habitat tolerances. Rather than using this temporal dimension to frame their analyses, I would suggest running the same sets of analyses for the two datasets and looking instead at how determinants of occurrence / richness have changed, and exploring how those differences might relate to the dramatic land use changes that have occurred in the intervening period.

We agree that trends in abundance are difficult to identify with data collected from only two survey periods and acknowledge this in the manuscript (e.g. lines 600-605). It is our hope that this study provides the impetus for a continued long-term monitoring effort.

We would not necessarily expect any difference in those factors underlying patch occupancy and abundance of birds in Tasmania between survey periods, perhaps only their relative strength. We had, nonetheless, considered repeating our multivariate statistical analyses on the original survey data as suggested by the reviewer. Unfortunately, the historical data was not recorded in such a way that the *mvabund* analysis could be used; bird data was recorded as densities and the raw abundance data was not available. We could have run the analysis for the two data sets using only presence/absence information but feel that this would not make use of the most interesting source of variation in the data.

- 2) While I appreciate the method used to estimate bird occurrence was effectively decided by MacDonald and Kirkpatrick for the 1996/97 survey period, many of the concerns I have with the data relate to how it has been handled, rather than initially collected. Variable effort was used which is not necessarily problematic. Rather, by adjusting their data based on proportion of the site surveyed, a systematic error forcing a species-area relationship might be the real problem. This can be avoided by using estimated richness rather than observed richness, following the methods of Colwell in his freely available EstimateS software. They can then look at how much the predicted richness diverges from observed richness and, if there are consistent effects of sampling effort, they can be quantified and more precisely dealt with. As for density, I have even more concern with the response variable used, and would recommend the authors follow the work of Mac Nally and others and uses incidence or reporting rate instead. That is, the proportion of surveys conducted in that site in which the species was detected as present. It is coarse, but far less prone to the many confounding effects of variable detectability, (between species, sites and seasons).

Transect surveys were proportional in their length to the size of the woodland surveyed and so reflect a proportional survey effort. If we were to standardise the length of transects across woodland remnants of different size, we would fail to account for the greater diversity of habitat types, and therefore birds, in larger patches of woodland. Our measure of species richness was equal to the accumulated total number of species from our seasonal surveys (now clarified on lines 280-281).

We only used species richness in one analysis. This was concerned with detecting changes in richness within woodlands and between survey periods and then identifying potential factors that may explain these changes (e.g. changes in woody veg cover & noisy miner density). Because we were only interested in *change in species richness* and we used the same bird survey techniques as in the earlier survey period, we do not think it necessary to calculate predicted richness in this instance. We are thankful to the reviewer, however, for introducing us to EstimateS; the software looks very useful!

Reporting rates are most appropriate for large data sets with more than two data points (i.e. not ours, as indicated by the reviewer in their first comment). Reporting rates have, for example, proven very useful for analysing data collected through citizen science projects but this metric requires all count data to be simplified to presence/absence. It is in our opinion that this would result in a loss of the most interesting variation in our data set.

Densities for some species did not change between the two survey periods (20 years apart), providing some indication that our comparisons are robust to differences in detectability between survey periods. After consulting with an ecological statistician, we decided that occupancy modelling, which accounts for differences in species detectability, was unnecessary. Our study sites were largely open habitat types, surveyed repeatedly and there are very few cryptic bird species in the region.

3) The analyses used were complex and, although well explained, the authors side-stepped what I took away as one of their main findings—the avifauna changed very little between sampling periods, and most of the factors measured had little explanatory power, either at the individual species scale or for variously-defined groups. There was also a very large number of potential relationships to explore—figure 4 alone is a distillation of approximately 600 univariate relationships. The determinants of species richness very murky, with more than half the variation unexplained. Yes, this is a negative result, but if that result remains using predicted richness where comparability across sites is assured, then the authors should stand by it, and discuss this lack of signal more explicitly. The fact that these site-based factors had so little explanatory power suggests that landscape scale factors might be more important, that inter-specific differences and the masking effects of simultaneous positive and negative relationships might conceal any assemblage-wide trajectories.

Changes in avifauna were very much dependent on the field site. Some woodlands experienced a dramatic change in species richness, but yes, for many sites there was little to no difference in this measure (Figure S2). We disagree, however, with the wider premise that there was little change in avifauna between survey periods. Some species such as cockatoos and ravens have significantly increased in number (reflected in Figures 5 & 6) while others appear to have declined (e.g. pardalotes, Fig. 5). Similarly, we do not feel that the determinants of change in species richness were murky, constituting a negative result. We modelled change in richness using just three predictor variables and achieved an adjusted R-squared value of 0.45. This is a good fit when considering such complex ecological interactions (see Møller & Jennions 2002).

We understand that by grouping species into assemblages based on diet, body size etc. this masked simultaneous positive and negative relationships and increased the associated confidence intervals in our analyses of change in bird density. This was, however, a key aim of the study – to test for common trends among groups of birds that share life history traits.

4) Related to point 3, I would suggest the authors be a little more circumspect about their inferences. In figure 5, temporal changes for 33 species are depicted of which more than half

have confidence intervals that overlap with zero. To me, this suggests that either most species aren't changing or the approach used to monitor them across this period is unable to discern any change. Likewise, in figure 6: just four of the 21 trait-defined groups exhibited significant changes in density: waterbirds, raptors, large birds and plant eaters. Rather than any trait-based influence, these four groups likely reflect the clear increases noted in swamp harriers and shelducks.

- 5) I would recommend the authors consider subdividing this large manuscript into three more narrowly-defined contributions. The noisy miner story might well be worth carving off and exploring on its own. [Oh, and please don't use the term "reverse keystone". Keystones are by definition rare in their communities, in terms of numbers of individuals / proportion of biomass-- the term "despotic" is far more appropriate]. The determinants of diversity in two different time periods would also stand alone, leaving perhaps a third manuscript considering what you've found overall, what appears to be important and not important in structuring these bird assemblages in a highly modified system, how you might expect birds to respond to planned restoration, and some guiding predictions to revisit in another twenty years. This is all up to you but by lumping all of this into one paper, not only does it become unwieldy, but you simply don't have the space to consider your findings. There's plenty of interesting comments made in passing about particular sites, land-use practices, positive and negative responses to shifting land use. Reworking the paper into more clearly defined narratives will give you the structure and space to explore these in more depth, and make better use of all the hard work that went into compiling all of this information.

We have given this suggestion some serious thought and acknowledge that the manuscript is quite lengthy. Nonetheless, we have decided to leave the paper as one for the following reasons.

First, relationships between noisy miners and Australian bird communities have been extremely well-studied at the habitat patch and landscape levels (see Maron et al. 2013, Thomson et al. 2015, Mortelliti et al. 2016, Simmonds et al. 2019). We do not provide the required level of novel insight on this topic to justify an addition to this comprehensive body of literature [I have removed reference to the term reverse keystone as requested].

Second, our data set is not strong enough to justify a stand-alone study of temporal comparisons in bird species presence or abundance, with only 33 sites where surveys were repeated 20 years apart and no intervening records.

Lastly, we believe the advantage of the manuscript as it currently stands is that our analysis of those environmental factors correlated with patterns of bird species presence and abundance is complemented by findings from the temporal comparison in bird data. The following comments were received from two reviewers of the PhD thesis that this manuscript draws from:

"What I liked most about it was the multi-pronged approach used to explore the different mechanisms behind patterns of bird occurrence in the Tasmanian Midlands, going well beyond a focus on describing landscape and habitat correlates. This is done too, though, and yields more insight than usual as it combines an analysis of contemporary patterns with a temporal comparison, and together these approaches paint a much richer picture of the conservation ecology of this bird assemblage."

"The [manuscript] is very comprehensive and having analysed very similar data to this in the past, I commend the student for doing an excellent job on data which can be quite challenging to analyse well."

We have further reduced the length of the manuscript as far as possible.

Reviewer: 2

Bain et al use bird surveys (past and present) to compare species and their traits with habitat characteristics in the Tasmanian midlands. Over all I think the paper is well written, the methods appropriate and well executed and the results interesting and well interpreted. I particularly like the use of the fourth corner approach (and the resulting figures) to addressing the question of how to link species occurrence data with complex habitat variation. My comments on are relatively minor, but I would suggest that the authors look out for over-long sentences in their revision, and attempt to clarify some parts of the methods as suggested below. The manuscript could be trimmed in some places to save words.

- 6) Methods – generally fine, but more detail is needed about the method used in the original surveys – given the time that has passed since the first surveys, it is important to understand the possible factors that could confound the results between the past and now. E.g. was observer error quantified and if not, then this assumption should be spelled out more (the first mention I see is at L428 which is a bit late). I see GB did all the field work in the contemporary surveys- who did them in the past? Is it possible to quantify detection error between the surveyors?

Added: Historical surveys were also completed by a single observer (MM).

Michael Macdonald (co-author) completed all the original surveys. We were not able to quantify observer error. The strip-transect survey method used in both survey periods is described on lines 160-165.

- 7) L157: presumably the orientation and location of transects was different due to change in vegetation cover (or not?) between survey periods? If yes, say so.

Transects location and orientation was the same between survey periods, even at sites where clearing had occurred. Cleared areas never overlapped with the location of the original transects. The only site where transects did differ in their location (Woodstock Lagoon) was excluded from all analyses of change in bird communities/species richness. This is stated on line 169.

- 8) L158: if I understand correctly, small patches received 1 * 100m survey transect and large ones received 4 * 200m transects back-to-back over 800m, plus an addition 20min/2ha? was the orientation random in large woodlands? Why you used 2 survey approaches (transects and then 2ha searches) – did you pool the data for analysis or were they used for different parts of the analysis? It's not clear to me how these different methods are used in analysis

Correct, these are good points. The orientation of transects in large woodlands either replicated that of the earlier survey or was randomly located but always began at the edge and was directed towards the woodland interior. We have added the following on line 162 to clarify this point: *These [transects] were placed randomly if their location had not already been established in the historical survey period.*

We have deleted the reference in our methods to 2ha surveys in woodlands because this was also a source of confusion for other reviewers and, ultimately, data collected in these concurrent surveys was not analysed in this manuscript. Mention of 2ha surveys is now restricted to a single sentence in the Results: *Seven additional species were detected and identified from calls offsite, eight more were observed during concurrent 2-hectare/20-minute bird surveys in woodlands (not analysed in this study, Table S2) and four were found only in either pastures or grasslands (Table S3).*

Additional 2ha surveys were conducted for two purposes. First, to provide data that was useful for Birdlife Australia (2ha-20mins is the standard survey method for their data base). Second, these were conducted with the hope of comparing detection probabilities of birds using different survey methods. That is, comparing species lists derived from 2ha searches in the middle of woodland sites to transect and acoustic surveys. We did not have time to analyse data from these surveys and as such have deleted the relevant lines from the manuscript.

9) L198: “as a proxy for land use intensity.”

Change accepted.

10) L204: specify here where you sourced miner density estimates – is this derived from your own models?

Miner densities were derived from the bird surveys conducted in the 2017 survey period. We have added the following: *Density of miners was determined from bird surveys as described above (see 2.3).*

11) L265: I understand this limitation of mvabund but think it would be helpful for readers if you provided some explanation of what this might mean for interpreting your potential results, especially since the large forest patches had quadruple the effort at transects twice as long as those in small patches

We contacted an ecological stats consultant regarding this matter to see if there was any workaround. It was explained to us that offsets (to account for differences in the area surveyed) are not able to be used only when using LASSO for model fitting and selection but that this method was still the most suitable for our data. We have added this detail to line 272. In addition, we now include the following...

While the trait analysis does not, therefore, account for differences in the area of each transect surveyed, the results remain informative. Observed traits were unrelated to woodland patch size and so this limitation is expected to have minimal influence.

12) L270: Clarify species richness – is that just the number of species per survey? Or number per site over the survey periods?

Added: *Here, species richness was the total number of native bird species recorded at a site over the duration of each survey period.*

13) Fig 4. It would help to have the environmental variable names on both the top and bottom for readability, but not critical

We agree but understand that the figure would no longer meet the formatting requirements of the journal (too large for the page).

14) L413-17: long and slightly clunky sentence. Reword.

Changed to: *The number of pivot irrigation systems increased dramatically between 1997 and 2017, from three irrigators within 1 km of two survey sites, to 33 irrigators at 12 survey sites. Yet, there was no relationship between the number of nearby irrigation systems and changes in species richness.*

Hopefully this is a little better.

15) L440 and L671: remove 'productive' – plantations are referred to in other ways too elsewhere in the paper - be consistent

We have removed "productive" from lines 447 & 680. We have left the term in the first instance (line 199) to help establish that commercial eucalypt plantations are separate from the small eucalypt plantings established for biodiversity or shelter purposes on farms.

16) L465: "Granivores and raptors appeared..."

Change accepted.

17) L468: re 'common in urban environments' either needs a citation or delete (relevance to the section seems tangential anyway)

The comment is relevant because it identifies the bird as a species associated with land use change, in this case urbanisation. This is a personal observation.

18) L611: flock sizes are results

These flock sizes were not recorded during our formal surveys, however, and so we did not want to conflate these with our more substantive survey data.

Reviewer: 3

A very good paper. See attached file for some comments.

I have copied the comments from the pdf and address them here.

This is an excellent paper on a subject of global interest. It is well written, and the literature view is very thorough for recent studies. The paper can be published with very little change. However, the following points could add some value:

19) L281, L473 & Figure captions: The term "response ratio" could imply a response to an identified process such as agricultural intensification, whereas in fact it just refers to a change over time between the two survey periods (with agricultural intensification as one of several possible causal factors). Would "percent change" be a less misleading term?

We understand the problem with this terminology but the log-response ratio (described on lines 317-325) is a well-established (Hedges et al. 1999) and commonly used metric in studies assessing change in species abundance (see Roberson et al. 2016 or Edgar & Barrett 2012 for examples) as

well as in meta-analyses. For this reason, we feel that it would cause more confusion if we were to change the term in our paper. "Response ratio" only describes the statistical metric used and is not meant to imply the cause of change. Percent change is a different measure of effect magnitude.

20) The trait-based approach is clearly useful, and quite similar to the guild approach used in various similar studies. The latter approach would enable conclusions to be made about changes in the abundance of whole guilds, e.g. nectarivores, canopy-foraging insectivores or small birds and large birds, rather than just saying that several species with those traits increased or decreased. It seems that calculations were made along these lines (Fig. 6) and perhaps more could be made of them in the text? I found only one reference to Fig. 6, and that just related to large herbivores such as ducks (line 465).

We tend to agree. Specifically, we have realised that the Abstract had failed to mention the results of our trait analysis altogether. We have rewritten the abstract to include some details of the trait analysis.

Fourth-corner modelling showed strong negative responses of aerial foragers and exotics to increasing woodland cover; arboreal foragers were positively associated with projective foliage cover; and small-bodied species were reduced by the presence of a hyperaggressive species of native honeyeater, the noisy miner (Manorina melanocephala).

Having said this, the large confidence intervals (noted by Reviewer 1) for most trait groups in Fig. 6 overlap zero. Therefore, we did not feel it appropriate to comment too much on those insignificant results.

21) The basic results (as shown in Figure 2) are similar to those from a study of fragmented forests in south-eastern Victoria (Loyn 1985, 1987), which showed the role of noisy miners, and gave estimates of likely declines in numbers of species over time. The study was repeated 22 years later, and the further declines were found to be less than predicted at a set of sites where eucalypt plantations had been established nearby (MacHunter et al 2006). It may be worth comparing these results.

We were not aware of these studies. They are very interesting but for the purposes of keeping the manuscript as short as possible we have chosen to add only some of the references suggested by the Reviewer (for example, Loyn et al 2007 in comment 32, Dow 1977 comment 26).

22) Title: Not sure that 20 years is long-term. Perhaps "Changes over 20 years...."?

We have changed the title such that it no longer includes "long-term".

23) L27-28: "have" not "has" (subject is plural, "levels"), one of just two typos noticed.

Thanks, change made.

24) L92-93. Is it possible to give a reference for the use of Bass Strait islands as stopover sites? Stephen Garnett et al. wrote a paper on this in the 1980s.

Unfortunately, I cannot find the paper suggested by the reviewer, only papers by Garnett on birds of the Torres Strait. I have added two appropriate references by Dingle (2004) and Chan (2001).

Chan K. (2001) Partial migration in Australian landbirds: A review. *Emu - Austral Ornithology* **101**:281-292.

Dingle H. (2004) The Australo-Papuan bird migration system: Another consequence of Wallace's Line. *Emu - Austral Ornithology* **104**:95-108.

25) Clarify which years were Tasmania's coldest winter (presumably 2017?) and warmest summer (2017-18?).

We have added the appropriate years.

26) Could also refer to the early work on this subject: Dow (1977) showed that Noisy Miners exclude other birds, and Loyn (1985) showed the importance of this process in fragmented forests in rural landscapes (in Victoria).

We have now included Dow (1977) among our references.

27) Interesting that Striated Fieldwrens were recorded at all (albeit offsite). In Victoria I have also found them in rural environments (in weedy young pine plantations and woody weeds in farmland) but quite rarely (and not for many years): they remain common in or near saltmarsh and sparsely treed heath, but otherwise rare in farmland away from the coast. [This is just a comment for interest: no changes needed unless you wish to say something about the habitat where they were found.]

Yes, we believe them to be quite rare on farmland in our study region. For your interest, we have observed striated fieldwren at two locations: near a very isolated single shrub in the centre of a bare paddock; and in another paddock where tall grasses and gorse shrubs were present.

28) L351-352: The paper by Maron et al (2013) mentioned that there was weak evidence for large-bodied birds benefitting from Noisy Miners (and strong evidence for small-bodied birds being disadvantaged, as already mentioned in this paper). It's good to see the Tasmanian study providing corroborative evidence with respect to two of the large-bodied species. (I've also found evidence for that, in recent studies in the Victorian mallee.)

29) L541-542: Similar shifts in bird communities have been observed in various studies. However, this particular example is interesting (and it is remarkable that the threshold fits so well with the predictions by Thompson et al.) so I think it's worth discussing, despite being based on just two sites.

Agreed. We wouldn't make so much of this point if our data did not seem to match so well!

30) L586-590: This last sentence may need rethinking. It could be misread as suggesting that some of Watson's other declining species actually do inhabit Tasmania (whereas the point is there may not have been enough monitoring to detect declines in the species known to be there). And is that really right? There are plenty of records in Birddata, which have been analysed at various times for Tasmanian bird reports and two recent reports on "the state of Tasmania's birds" (Newman et al. 2017, 2019). Perhaps it's better to just say that this points to the need for more monitoring in Tasmania, in a range of environments including rural and forested landscapes, and then make a considered comment on what the current data tell us about resilience. (There have been recent suggestions that several Tasmanian non-endemic species may be declining, including Blue-winged Parrot and Flame Robin: it's pleasing to see that the current dataset suggests the reverse for Bluewinged Parrot, and little change for Flame Robin.)

We have rewritten this sentence to avoid misinterpretation.

31) L614: Did you class corellas as introduced species? I've tended to think of them as natural recent colonists (as they are in southern Victoria), and this is supported by the first record of Little Corella being in 1983 (a major drought year in SE Australia). However, I notice they are listed as introduced to Tasmania in the State of Tasmanian Birds reports: could be worth checking.

The Tasmanian government list corellas as an introduced species, it was for this reason that we also classed them as introduced. This is noted in Table 2, paragraph 5 of the Results and in the Discussion. We cannot definitively say whether corellas were self-introduced to Tasmania or not (much like debate over other parrots such as galahs in Launceston).

32) L635-665: A study in Victoria found that Noisy Miners rarely use young commercial or amenity eucalypt plantations (Loyn et al. 2007), though parallel work found that they were often present in mature linear planted shelterbelts.

L665: A positive effect of new eucalypt plantations was also observed by MacHunter et al. (2006).

L656: There were two papers by Grey et al. reporting removal experiments on different sets of sites. The second paper is Grey et al. (1998). There have also been several removal experiments with Bell Miners, with similar results.

Thanks for the references, they will be useful in future. We now cite the Loyn (2007) study but are aware that we need to keep the manuscript short, and so have been quite selective in the references we choose.

33) L689: the second typo, add apostrophe ("weed's).

Corrected.

References (if not already in text)...

No need to include them all, but some could be useful for this or subsequent papers...

Many thanks to the reviewer for their effort in sharing these references with us. Very helpful!

Catterall, C.P., Kingston, M.B. and Park, K. 1997 Use of remnant forest habitat by birds during winter in subtropical Australia: patterns and processes. *Pacific Conservation Biology* 3:262-74

Catterall, C.P., Kingston, M.B., Park, K. and Sewell, S. 1998 Effects of clearing lowland eucalypt forests on a regional bird assemblage. *Biological Conservation* 84:65-81

Dow, D.D. 1977 Indiscriminate interspecific aggression leading to almost sole occupancy of space by a single species of bird. *Emu* 77:115-21.

Grey, M.J., Clarke, M.F. and Loyn, R.H. 1998. Influence of the Noisy Miner *Manorina melanocephala* on avian diversity and abundance in remnant Grey Box woodland. *Pacific Conservation Biology* 4: 55-69.

Loyn, R.H. 1985. Birds in fragmented forests in Gippsland, Victoria. Chapter 31 in Keast, A., Recher, H.F., Ford, H.A. and Saunders, D. (eds), pp 324-331. *Birds of the Eucalypt Forests and Woodlands: ecology, conservation and management*. Surrey-Beatty, Sydney.

Loyn, R.H. 1987. Effects of patch area and habitat on bird abundances, species numbers and tree

health in fragmented Victorian forests. Chapter 6 in Saunders, D.A., Arnold, G.W., Burbidge, A.A., and Hopkins, A.J.M. (eds), *Nature Conservation: the role of remnants of native vegetation*, pp. 65-77. Surrey Beatty & Sons, Sydney, 410 pp.

Loyn, R.H., McNabb, E.G., Macak, P. and Noble, P. 2007. Eucalypt plantations as habitat for birds on previously cleared farmland in south-eastern Australia. *Biological Conservation* 137: 533-548. Doi:10.1016/j.biocon.2007.03.012.

MacHunter, J., Wright, W., Loyn, R. and Rayment, P. 2006. Bird declines over 22 years in forest remnants in south-eastern Australia: evidence of faunal relaxation? *Canadian Journal of Forest Science* 36: 2756-2768.

Newman, M., Ramshaw, N., Walter, A., Webber, W., Drake, S. and Woehler, E. 2017. The state of Tasmania's terrestrial birds 2014-15. Birdlife Tasmania.

Newman, M., Ramshaw, N., Drake, S., Woehler, E., Walter, A. and Webber, W., 2019. The state of Tasmania's birds 2015-16. Birdlife Tasmania.

References

Møller A.P. and Jennions M.D. (2002) How much variance can be explained by ecologists and evolutionary biologists? *Oecologia* **132**(4): 492-500.

Hedges L.V., Gurevitch J. and Curtis P.S. (1999) The meta-analysis of response ratios in experimental ecology. *Ecology* **80**(4): 1150-1156.

Edgar G.J. and Barrett N.S. (2012) An assessment of population responses of common inshore fishes and invertebrates following declaration of five Australian marine protected areas. *Environmental Conservation* **39**(3): 271-281.

Roberson E.J., Chips M.J., Carson W.P. and Rooney T.P. (2016) Deer herbivory reduces web-building spider abundance by simplifying forest vegetation structure. *PeerJ* 4:e2538
<https://doi.org/10.7717/peerj.2538>

Appendix - Responses to additional comments made by thesis examiners since original manuscript submission.

- 1) [Thesis Rev 1] This is a good solid chapter describing in detail patterns of woodland bird occurrence and change across the Tasmanian Midlands. I hope to see it published soon. I feel that the abstract potentially be a little clearer about the design and the practical recommendations, but overall it is a nice piece of work, well done!

I have altered the final line of the Abstract to give a specific recommendation for restoration practitioners.

Now: We encourage restoration practitioners to trial novel planting configurations that could be more resilient to invasion by noisy miners as well as a continued long-term monitoring effort such that the effects of future landscape change (revegetation or continued habitat loss) on Tasmanian bird communities can be examined.

- 2) [Thesis Rev 1] L 1295 – can you be more precise? How many species are we talking about here?

Added a reference and estimate of the number of migratory species (19+). Previous researchers have included species that I consider only vagrants to Tasmania. It is, therefore, difficult to settle on any specific number.

- 3) [Thesis Rev 1] L 1300 – explain ‘reverse keystone’

Added explanation and citation: *Agricultural fragmentation in the Midlands has also led to the dominance of a reverse keystone species, the noisy miner (Manorina melanocephala), in many small patches of habitat (MacDonald & Kirkpatrick 2003); for a location to support a typical, diverse woodland bird community, noisy miners must be absent or at very low density (Piper & Catterall, 2003).*

- 4) [Thesis Rev 1] L 1306 – is there evidence that they are a ‘major predatory threat’ in this landscape – as in, enough to affect populations?

I have added an additional reference that highlights feral cats as a major predatory threat to birdlife throughout Australia. It is, therefore, appropriate to expect the same of feral cats in the Midlands region.

Also added qualifier: *... feral cats which are a major predatory threat to birdlife, although it is unknown how mortality from cat predation might translate to the population viability of birds (Hamer 2019).*

- 5) [Thesis Rev 1] L 1312 – careful with inadvertent tense changes

Now: Second, we explored changes in the Midlands bird community over the past twenty years as a consequence of land-use change. The only broad-scale survey of birds in the Tasmanian Midlands was previously conducted in 1996-98 (hereafter the 1997 survey period, MacDonald & Kirkpatrick 2003). We repeated surveys at 34 historical study sites to explore the effects of habitat clearing, revegetation and changes in noisy miner abundance on bird communities. Changes in bird abundance between survey periods allowed for the identification of target species and functional groups of birds to be either encouraged or discouraged by restoration efforts.

- 6) [Thesis Rev 1] L 1352- what are these connections? Actual or theoretical, planned, movements of birds or places where trees will be planted? Can you elaborate?

With reference to Figure 3.1 and the statement: *Arrows signify the conceptual east-west connections of the Midlands restoration program.*

Now: Arrows signify the conceptual east-west connections of the Midlands restoration program where revegetation is planned to occur. The “Ross connection” is currently being established with some sections of this corridor already five years of age.

- 7) [Thesis Rev 1] L 1354 onward – the methods section is hard to follow and the design does not come through clearly. There are sites, plots, transects, and remnants/remnant woodlands. I am not sure if there were multiple sites within remnants and multiple plots per site, or if sites and plots are the same, etc. Suggest a rewrite with attention to this issue.

I have removed any reference to “plots” when describing the collection of bird survey data. This term is now reserved for the first paragraph of “Environmental Data”, in which I explain methods

of collecting vegetation data. The term “transects” is only used in reference to bird surveys. I have removed “sampling points” from the caption of Table 3.1, replacing with the “centre of bird survey transects”.

8) [Thesis Rev 1] L 1418 – can you describe what TASVEG is?

Added: TASVEG is a state government digital map that depicts the extent of 162 vegetation communities in Tasmania (DPIPWE 2013). We updated the spatial layer to more accurately reflect vegetation composition at our survey sites, for example, where recent land clearing had occurred.

9) [Thesis Rev 1] Table 3.1 mentions sampling points – are these different to plots/transects etc

I have changed “sampling points” to “the middle of transects” in the table caption.

10) [Thesis Rev 1] L 1449 – but most are not behaviours

With reference to: These traits were body size, diet, foraging height, native status and movement pattern (Table 3.2). We acknowledge that many species do not exclusively fit one category type for each trait, but for the purposes of analysis, birds were categorised according to their primary behaviours.

Now: ... according to their primary modes.

11) [Thesis Rev 1] L 1467 – why not use residual patch shape after accounting for area?

I agree, this is something I realised only after completing the analysis. I have run the *mvabund* analysis again using the following formula from Robertson *et al.* (2014) to account for area as suggested:

$$\text{Corrected perimeter:area ratio} = \frac{\text{perimeter (m)}}{\sqrt{4\pi \times \text{area of patch}}}$$

This has changed my results and has required a rewrite of some sections (e.g. *Relationships between environmental variables and the bird community*) as well as redrawing Figures 3.3 & 3.4 and updating the results in Table 3.3. The new analysis now identifies Landcover diversity and Leaf litter cover as having significant effects on community composition of birds at survey sites. The order of significance of other environmental variables is also different but the overall interpretation of my data has not changed dramatically.

12) [Thesis Rev 1] L 1651 – density is per unit area – indicate this (is it per ha?)

Correct. I have added the appropriate units (hectare⁻¹).

13) [Thesis Rev 1] L 1772 – what do you mean ‘should be avoided’ – can you explain how?

*Was: Thus, such dramatic changes in community composition are consistent with the impact threshold proposed by Thomson *et al.* (2015) and should be avoided in restored habitat.*

*Now: Thus, such dramatic changes in community composition are consistent with the impact threshold proposed by Thomson *et al.* (2015); this represents an effective target for the maximum density of noisy miners that could be tolerated by other birds in restored habitats.*

14) [Thesis Rev 1] L 1872 – these were both short-term studies – not long-term

Correct. I have changed “long-term” to “short-term”.

15) [Thesis Rev 1] L 1886 – presumably only relative to some specific matrix types – specify

Was: Law et al. (2014) found that as eucalypt plantations in northern New South Wales matured, abundance of noisy miners declined.

I have specified: ... in a farmland mosaic of northern New South Wales...

16) [Thesis Rev 1] L 1891 – what is implied by ‘restoring’ here?

Was: Huth and Possingham (2011) showed that there is greater benefit to avian biodiversity in restoring small and degraded woodlands than there is in increasing their size.

Added: ... in restoring structural attributes small and degraded woodlands than there is in increasing their size.

17) [Thesis Rev 2] L 1399 I would delete this paragraph - if these are not included in this chapter then there is no need to include them in the methods section

See 46. I will of course adopt this advice and delete the paragraph when it comes to publication of the chapter.

18) [Thesis Rev 2] Fig 3.4 - I was disappointed to see the species list alphabetically - it would be far more interesting to see these listed by foraging groups or by those which shared similar relationships - something (anything) would be more revealing than having them being listed alphabetically.

This is a great suggestion. I have redone this analysis and ordered the species by foraging groups.

19) [Thesis Rev 2] L 1657 what does this mean? This is a very (too) broad a statement - and is not accompanied by any statistics. Do you mean that it had changed significantly - or just that it wasn't identical? and when you say change - does that include both increases and decreases? Perhaps also state out of how many sites in total? 33? I am confused why this is in a new paragraph - I think this should be linked to the previous paragraph?

Was: Woody vegetation cover had changed at 14 sites. At 11 of these, agricultural pastures and exotic vegetation had replaced native eucalypt woodland and at one site native woodland was replaced by both open water (farm dam) and pasture. The average loss of woody vegetation cover was 3% (range = 1 to 8%).

Two small remnants experienced an increase in surrounding woody vegetation cover, due to the establishment of a pine plantation near one site (+17%) and a productive eucalypt plantation at the other (+21%).

Restructured: Woody vegetation cover had changed at 14/33 sites. At 12 of these, woodland cover had declined with an average loss of 3% (range = 1 to 8%). Native woodland was replaced by agricultural pastures and exotic vegetation at 11 sites and by both open water (farm dam) and pasture at one location. Two small remnants experienced an increase in surrounding woody vegetation cover due to the establishment of a pine plantation near one site (+17%) and a productive eucalypt plantation at the other (+21%).

I mean that it wasn't identical. The sentences immediately following explain the direction and magnitude of the change in woodland cover. Statistical analyses are unnecessary here. I have

added "14/33 sites" and joined these paragraphs as suggested plus restructured the section slightly.

20) [Thesis Rev 2] where was the plot place in the woodland patch and how was this decided upon - randomly? what about in relation to the edge of the patch?

Now: *At the centre of woodland sites and plantings, we established two intersecting 10 m x 50 m plots and recorded the following site-scale attributes... Plots were positioned on survey transects at the centre of woodlands and plantings to accommodate concurrent research on habitat use by mammals [29, 34].*

21) [Thesis Rev 2] L 1418 where was the centre of this buffer located - or was it a buffered around the wooldand patch itself - and thus not a circle - please clarify?

Was: *Satellite data and the geospatial layer, TASVEG 3.0 (updated), were used to classify landcover within a 1 km radius of survey locations (DPIPWE 2013).*

Now: *Satellite data and the geospatial layer, TASVEG 3.0 (DPIPWE 2013), were used to classify landcover within a circular buffer zone of one-kilometre radius that was centred on the middle of bird survey transects.*

22) [Thesis Rev 2] L 1436 not clear which transect this refers too?

Was: *Values for FPC were averaged over the area of each transect.*

Now: *Values for FPC were averaged over the area of each bird survey transect.*

23) [Thesis Rev 2] L 3806 where? in them or surrounding them?

With reference to: *Typically, small and degraded patches of woodland were characterised by exotic pasture grasses, bracken fern (*Pteridium esculentum*) and introduced weeds including gorse and hawthorn (*Crataegus monogyna*).*

Added: *Typically, small and degraded patches of woodland were characterised by exotic pasture grasses throughout their interior, bracken fern (*Pteridium esculentum*) and introduced weeds including gorse and hawthorn (*Crataegus monogyna*).*

Appendix C

- 1 **Long-term change in Changing bird communities of an agricultural landscape:**
- 2 **declines in arboreal foragers, increases in large species.**

Glen C Bain^{*a}, Michael MacDonald^b, Rowena Hamer^c, Riana Gardiner^d, Chris N Johnson^e and Menna E Jones^f.

^a *School of Natural Sciences, University of Tasmania, Private Bag 55, Hobart TAS 7005, AUSTRALIA*

* Corresponding author email: glen.bain@utas.edu.au, phone: +61 405063230, Orcid ID: 0000-0003-1760-6706

^b *RSPB Centre for Conservation Science, RSPB Cymru, Castlebridge 3, 5-19 Cowbridge Road East, Cardiff, CF11 9AB, UNITED KINGDOM*

Email: michael.macdonald@rspb.org.uk

Orcid ID: 0000-0002-1061-8504

^c *School of Natural Sciences, University of Tasmania, Hobart TAS 7005, AUSTRALIA*

Email: rowena.hamer@utas.edu.au

Orcid ID: 0000-0002-9063-5426

^d *School of Natural Sciences, University of Tasmania, Hobart TAS 7005, AUSTRALIA*

Email: riana.gardiner@utas.edu.au

Orcid ID: 0000-0001-7782-8664

^e *School of Natural Sciences, University of Tasmania, Hobart TAS 7005, AUSTRALIA*

Email: c.n.johnson@utas.edu.au

Orcid ID: 0000-0002-9719-3771

^f *School of Natural Sciences, University of Tasmania, Hobart TAS 7005, AUSTRALIA*

Email: menna.jones@utas.edu.au

Orcid ID: 0000-0001-7558-9022

**Keywords**

- Agricultural intensification, avian declines, ecological restoration, land-use
- change, life-history traits, woodland birds

Abstract

~~Populations of many b~~Birds are declining in agricultural landscapes around the
~~world. Mechanisms underlying~~The causes of these declines ~~can be better~~can be
~~better understood by exploring population~~analysing change in groups of species that
~~share life-history traits. We investigated how land-use change has affected birds of~~
~~the Tasmanian Midlands, one of Australia's oldest agricultural landscapes and a~~
~~focus of habitat restoration and the focus of an ambitious habitat restoration~~
~~program.~~ We surveyed birds at 72 sites, some of which were previously surveyed in
~~1996-1998, and tested relationships of current patterns of abundance and~~
~~community composition to landscape and patch-level environmental characteristics.~~
~~Fourth-corner modelling showed strong negative responses of Woodland cover, at~~
~~survey sites, projective foliage cover, densities of a hyperaggressive honeyeater (the~~
~~noisy miner *Manorina melanocephala*), landcover diversity, elevation and leaf litter~~
~~cover all had significant their elevation, and densities of a hyperaggressive~~
~~honeyeater, the noisy miner (*Manorina melanocephala*), had the strongest effects~~
~~on the composition of local bird communities. Fourth-corner modelling a showed~~
~~strong negative responses of aerial foragers and exotic birds to increasing woodland~~
~~cover; a. Arboreal foragers were positively associated with projective foliage cover;~~
~~and small-bodied species were most negatively influenced~~reduced by the presence
~~of a hyperaggressive species of native honeyeater, the noisy miner (*Manorina*~~
~~*melanocephala*). Analysis of change suggests an-increases in large-bodied~~
~~birdsgranivorous or carnivorous with a granivorous or carnivorous diets~~birds, and
~~but declines in some arboreal foragers and nectarivores. Common species, including~~
~~Tasmanian endemics, were among those with the largest proportional declines.~~

Changes in species richness were best explained by changes in noisy miner
abundance and levels of surrounding woodland cover. We ~~make~~ encourage
restoration practitioners to trial novel planting configurations that may ~~be~~
resilient confer resistance to invasion by noisy miners, and practical
recommendations to restore habitat for terrestrial birds of Tasmania and encourage
a continued long-term monitoring effort ~~such that the~~ to reveal effects of future
land-use change on Tasmanian birds ~~can be examined~~.

1. Introduction

Agricultural intensification is a major cause of global biodiversity loss [1]. Beyond the
direct clearing and fragmentation of habitat that typically accompanies agriculture,
farm management practices can modify wildlife communities through their effects
on the abundance of food, shelter and predators. For example, rising levels of
pesticide use to protect crops from herbivory haves had wide-ranging negative
impacts on invertebrate populations [2] and haves subsequently been implicated as
a major contributor in the decline of farmland bird populations in Europe and North
America [3-5]. In contrast, corvids and mammalian mesopredators (for example feral
cats *Felis catus* and raccoons *Procyon lotor*) are thought to have benefitted from an
increased food supply on farms and fewer large predators [6, 7]. The mechanisation
of agriculture and expansion of irrigated lands also results in the removal of discrete
habitat elements such as paddock trees, surface rocks, coarse woody debris and
hedgerows [8, 9]. These features are often the only source of cover for animals in
farm landscapes and are used by a range of taxa for breeding, thermoregulation and
as sources of food [10-12].

Understanding how land use affects wildlife communities is essential if we are to
predict their response to future environmental change, whether that results from
continued agricultural intensification or landscape restoration. This is especially
important given that the agricultural sectors need to meet a projected increase in
global food demand of between 25 and 70% [13]. Land-use change associated with
agriculture can affect species differently according to their life-history traits such as
body size, diet or dispersal ability. Relationships between functional traits of species
and population change can offer insight into the mechanisms by which land use

affects wildlife [14, 15]. This understanding can be applied to produce more targeted
conservation strategies and may allow for broader approaches that protect several
species at once [16, 17]. Functional traits can also be used to estimate the
contribution of species to ecosystem services and, therefore, how those services
might respond to land-use change [18].

Birds have long been used as indicators of environmental condition and can be
effective indicator species for the conservation of other threatened vertebrates [19].
They occupy many ecological niches, respond relatively rapidly to changes in habitat,
are straightforward to monitor, and perform important ecological functions such as
seed dispersal, pest control and pollination [20]. We surveyed birds in the Tasmanian
Midlands, one of Australia's oldest agricultural landscapes, a National Biodiversity
Hotspot and the focus of an ambitious habitat restoration program [21]. The bird
community of the Midlands has high biodiversity value, with ten Tasmanian endemic
species and several distinct subspecies (Table S1). Tasmania, more generally, is
important for many species of terrestrial bird (at least 19) that migrate to the island
from mainland Australia to breed each year [22, 23].

An increasing threat to local bird communities is secondary clearing of remnant
woodlands for the installation of large pivot irrigation systems [24]. Agricultural
fragmentation in the Midlands has also led to the dominance of an aggressive
species of native honeyeater ~~reverse keystone species~~, the noisy miner (*Manorina*
*melanocephala*), in many small patches of habitat [25]. Noisy miners are native to
Australia's eastern coast but have become overabundant [26]. The noisy ~~if~~
~~aggressive exclusion~~ miner's exclusion of small birds from suitable woodland habitat

is listed as a *Key Threatening Process* under federal environment legislation [27]. The
productive landscapes of the Midlands also support very high densities of feral cats
which are a major predatory threat to birdlife [28], although it is unknown how
mortality from cat predation might translate to population viability of birds.

Our first aim was to ~~identify~~ assess changes in the Midlands bird community over the
past twenty years as a consequence of ~~significant changes in~~ land-use change. The
only broad-scale survey of birds in the Tasmanian Midlands was previously
conducted in 1996-98 [hereafter the 1997 survey period, 25]. We repeated surveys
at 34 historical study sites in order to explore the effects of habitat clearing,
revegetation and changes in noisy miner abundance on bird communities. Changes
in bird abundance between survey periods allowed for the identification of target
species and functional groups of birds to be either encouraged or discouraged by
restoration efforts. ~~This will also help to identify particular species or functional~~

[revised manuscript text omitted]

~~of each transect survey, we also conducted a 2-hectare/20-minute bird survey [32] in~~
~~the centre of each woodland.~~

2.4 Environmental Data

At ~~the centre of~~ woodland sites and plantings, we established two intersecting 10 m
x 50 m plots and recorded the following site-scale attributes: the number of alive
and dead stems (> 10 cm diameter at breast height, DBH), the DBH of alive and dead
stems (cm) from which basal area was later calculated (m²), average canopy height
(m) and the summative length of fallen logs (> 10 cm in diameter, m). For analysis,
these data were extrapolated beyond the sampling plot to be expressed per hectare.
Plots were positioned on survey transects in the centre of woodlands and plantings
to accommodate concurrent research on habitat use by mammals [28, 33]. A line-
point intercept method (2 x 50 m transects) was used to estimate the percent cover
of ground substrates which were classified into eight categories: leaf litter, bare
earth, rocks, forbs and other low herbaceous plants, shrubs (> 30 cm in height,
including bracken fern), sedges, mosses and lichen. Shannon's Diversity Index (H')

was calculated to represent heterogeneity of groundcover [34].

We used the software program ArcMap [version 10.4.1, 35] to calculate the size (ha)
and shape ~~complexity (perimeter/area)~~ of woodland remnants. Shape was calculated
as the corrected perimeter to area ratio using the formula: $shape = \frac{perimeter\ of\ woodland}{\sqrt{4\pi(Area\ of\ woodland)}}$
. Satellite data and the geospatial layer, TASVEG 3.0-[36], were

used to classify landcover within a buffer zone of one-kilometre radius that was
 centred on the middle of bird survey transects. ~~of survey locations.~~ TASVEG is a state
 government digital map that depicts the extent of 162 vegetation communities in
 Tasmania [36]. We updated the spatial layer to more accurately reflect vegetation
 composition at our survey sites, for example, where recent land clearing had
 occurred. For historical survey sites, patch size and landcover were also determined
 from digitised aerial images collected in 1997 to allow calculation of change between
 the 1997 and 2017 surveys. Seven categories of landcover were identified: (1) native
 eucalypt woodlands, including mixed eucalypt plantings; (2) native non-eucalypt
 woodlands; (3) productive eucalypt plantations; (4) pine (*Pinus radiata*) plantations;
 (5) agricultural pastures and other exotic vegetation; (6) open water (mostly farm
 dams); and (7) native grasslands (Table 1). Categories 1-4 were further classified as
 “woody vegetation”. Cover of native grasslands could not be determined from aerial
 imagery, so for all analyses, the distribution of grasslands in 1997 was assumed to be
 unchanged from that in 2017. Shannon’s Diversity Index was again used to represent
 heterogeneity of landcover.

Table 1. Summary of landcover types within 1 km of ~~sampling points~~the centre of
 bird survey transects at woodland ($n = 52$) and planting sites ($n = 5$).

Land Cover Class	Mean % (standard deviation)	Min %	Max %
Eucalypt woodlands	35.3 (26.6)	0.2	96.5
Non-eucalypt woodlands	0.8 (0.4)	0.0	18.9
Eucalypt plantation	0.4 (2.7)	0.0	20.7
Pine plantation	0.4 (0.3)	0.0	17.0
Production agriculture	54.8 (3.8)	0.0	99.1
Native grassland	6.3 (1.6)	0.0	56.5
Open water	2.2 (0.6)	0.0	26.2

We counted the number of centre-pivot irrigation systems present within one
kilometre of survey sites as a proxy for ~~intensive~~-land use intensity. Pivot irrigators
need relatively flat terrain to operate and usually require large areas of land (up to
300 ha) to be entirely cleared of native vegetation. Climate data (mean annual
rainfall, rainfall seasonality, mean annual temperature), as well as the elevation of
survey sites and projective cover of woody foliage (Foliage Projective Cover, FPC)
were derived from geospatial information systems [37, 38]. Values for FPC were
averaged over the area of each bird survey transect. Finally, the density of noisy
miners at survey sites was considered as an environmental variable because
previous research has demonstrated strong effects of noisy miners on the
abundance of other bird species [39]. Density of miners was determined from bird
surveys as described above (see 2.3).

*2.5 Trait Data*

All birds observed were categorised according to species traits (Tables S1-S3) using
data extracted from the Handbook of Australian, New Zealand and Antarctic Birds
[40-46]. These traits were body size, diet, foraging height, native status, and
movement pattern (Table 2). We acknowledge that many species do not exclusively
fit one category type for each trait, but for the purposes of analysis, birds were
categorised according to their primary behaviours.

**Table 2.** Summary of trait variables used to explain variation across bird species in
their response to environmental variables.

Trait	Category	Example Species
-------	----------	-----------------

Body Size	Very Large (> 1000 g)	wedge-tailed eagle, Australian shelduck
	Large (100-1000 g)	sulphur-crested cockatoo, grey currawong
	Medium (25-100 g)	eastern rosella, fan-tailed cuckoo
	Small (< 25 g)	scarlet robin, superb fairy-wren
Diet	Invertebrates	Australian magpie, welcome swallow
	Vertebrates	brown falcon, nankeen kestrel
	Seeds & Grains	galah, house sparrow
	Nectar	musk lorikeet, eastern spinebill
	Plants	grey teal, Australian shelduck
	Generalists	black currawong, forest raven
	Foraging Height	Terrestrial
Height	Arboreal	striated pardalote, little wattlebird
	Arboreal / Terrestrial	grey-shrike thrush, grey butcherbird
	Aerial	welcome swallow, brown falcon
Movement Pattern	Resident / Sedentary	yellow-throated honeyeater, grey butcherbird
	Migratory *	golden-whistler
	Nomadic †	silveryeye, dusky woodswallow
Native Status	Native to Tasmania	crescent honeyeater, musk lorikeet
	Exotic to Australia	green rosella, spotted pardalote
	Introduced to Tasmania	common starling, European goldfinch laughing kookaburra, little corella

* Migratory species included only those birds known to migrate to mainland Australia
 annually, including partial migrants. † Nomadic species also included altitudinal
 migrants and species described as dispersive in the literature.

2.6 Statistical Analyses

Our analysis was performed in five stages. First, we used the package *BORAL* in R to
 produce a model-based unconstrained ordination of bird-count data for the
 visualisation of species-site relationships [47, 48]. In all analyses of multivariate data,
 we pooled observations of birds from transect surveys at each site and specified a
 negative binomial distribution. We included a fixed row effect when generating our
 ordination to account for differences in the total abundance of birds at survey sites
 such that the resulting ordination reflects species composition.

Next, we used the package *mvabund* (version 3.13.1) to test for effects of
 environmental variables on bird community composition at survey sites [49].

Candidate environmental variables were reduced to a final set of 10 (Table 3) by
generating a correlation matrix (Spearman's rank) and eliminating those variables
that demonstrated high levels of collinearity ($r_s > 0.6$). ~~One exception was that we~~
~~included both patch shape and woody vegetation cover ($r_s = 0.78$) in our models. We~~
~~did this for three reasons: bird species respond differently to each of these habitat~~
~~characteristics; both were considered to be biologically informative; and removing~~
~~either variable would lead to biased estimates of importance for other predictors.~~
~~Patch shape was also strongly associated with patch size ($r_s = 0.94$), as was the cover~~

[revised manuscript text omitted]

diversity, elevation FPC and the percent cover of leaf litter. Basal area and landcover
diversity also had some minor influence on the bird community ($p = 0.070$ & 0.075 ,
respectively). Fifteen species responded significantly ($p < 0.05$) to increasing cover of
woody vegetation. Of these, six responded negatively and nine species, including
three that are endemic to Tasmania, responded positively (Table 3). Univariate tests
showed that no individual species contributed significantly to the effects of FPC and
the cover of leaf litter on the multivariate bird community. Increasing density of
noisy miners negatively affected seven small-bodied birds (< 25 g) but was positively
associated with two species, the Australian magpie (*Cracticus tibicen*) and eastern
rosella (*Platycercus eximius*). Despite elevation having a strong effect at the
community level, the grey fantail (*Rhipidura albiscapa*) was the only species to
respond significantly negatively in univariate tests (Table 3), although black-headed
honeyeaters (*Melithreptus affinis*) tended to be more common in woodlands at
higher elevations ($p = 0.072$).

**Table 3.3.** Summary of a multivariate analysis (*manyGLM*) testing for the effects of
environmental variables on bird community composition. *P*-values (< 0.05 bolded)
and Wald statistics are given for the effect of variables at the community level.
Estimates ± standard error are for individual species that contributed significantly to
the variance in community composition. The sign of the estimate (positive or
negative) indicates the direction of a species response to the environmental variable.

Environmental Variable	Wald (1 df)	Community p value	Species where p < 0.05	estimate ± se
Woody Vegetation Cover	20.95	0.001	Australian magpie (Cracticus tibicen)	-0.015 ± 0.007
			common starling (Sturnus vulgaris)	-0.029 ± 0.006
			eastern rosella (Platycercus eximius)	-0.031 ± 0.012
			European goldfinch (Carduelis carduelis)	-0.037 ± 0.008
			forest raven (Corvus tasmanicus)	-0.010 ± 0.005
			grey butcherbird (Cracticus torquatus)	-0.019 ± 0.006
			black-headed honeyeater (Melithreptus affinis)	+0.053 ± 0.048
			crescent honeyeater (Phylidonyris pyrrhopterus)	+0.034 ± 0.013
			eastern spinebill (Acanthorhynchus tenuirostris)	+0.095 ± 0.033
			fan-tailed cuckoo (Cacomantis flabelliformis)	+0.058 ± 0.034
			grey currawong (Strepera versicolor)	+0.039 ± 0.013
			grey shrike-thrush (Colluricincla harmonica)	+0.064 ± 0.017
			scarlet robin (Petroica boodang)	+0.020 ± 0.006
yellow wattlebird (Anthochaera paradoxa)	+0.044 ± 0.008			
yellow-throated honeyeater (Lichenostomus flavicollis)	+0.013 ± 0.010			
Foliage Projective Cover	13.59	0.011	=	=
Miner Density	21.03	0.012	brown thornbill (Acanthiza pusilla)	-1.033 ± 0.189
			European goldfinch (Carduelis carduelis)	-1.394 ± 0.238
			grey fantail (Rhipidura albiscapa)	-0.952 ± 0.179
			scarlet robin (Petroica boodang)	-1.481 ± 0.378
			silvereye (Zosterops lateralis)	-3.096 ± 0.683
			superb fairy-wren (Malurus cyaneus)	-1.033 ± 0.222
			yellow-rumped thornbill (Acanthiza chrysorrhoa)	-2.590 ± 0.504
			Australian magpie (Cracticus tibicen)	+0.865 ± 0.129
eastern rosella (Platycercus eximius)	+1.738 ± 0.217			
Landcover Diversity H'	11.56	0.013	=	=
Elevation	13.65	0.014	grey fantail (Rhipidura albiscapa)	-0.006 ± 0.001
Leaf Litter Cover	10.61	0.039	=	=
Basal Area	9.95	0.088	=	=
Shape complexity	12.23	0.118	=	=
Canopy Height	9.57	0.468	=	=
Groundcover Diversity H'	8.74	0.546	=	=

~~Despite elevation having a strong effect at the community level, the grey fantail~~
~~(*Rhipidura albiscapa*) was the only species to respond significantly negatively in~~
~~univariate tests (Table 3), although black-headed honeyeaters (*Melithreptus affinis*)~~
~~tended to be more common in woodlands at higher elevations ($p = 0.072$). Increasing~~
~~density of noisy miners had a significant negative effect for seven small-bodied birds~~
~~(< 25 g) and was positively associated with two species, the Australian magpie~~
~~(*Cracticus tibicen*) and eastern rosella (*Platycercus eximius*). Univariate tests showed~~
~~that no individual species contributed significantly to the effects of FPC and the~~
~~cover of leaf litter on the multivariate bird community.~~

Life-history traits of birds explained their response to environmental variables. The
negative influence of noisy miners on small birds was once again highlighted by our
trait analysis (Fig. 3). Large birds showed no response to miner density but were
more common at lower elevations and at sites with a greater diversity of
surrounding landcover types. The positive association between miners and birds
introduced to Tasmania reflects a higher abundance of little and long-billed corellas
(*Cacatua sanguinea*, *Cacatua tenuirostris*) and laughing kookaburras (*Dacelo*
*novaeguineae*) in miner-dominated woodlands (Fig. 3). These three species have
been introduced to Tasmania from elsewhere in Australia. However, species exotic
to Australia were negatively associated with miners. Exotic species also showed a
strong negative response to increasing cover of woody vegetation and were more
common ~~in woodlands with a greater perimeter-area ratio (i.e. shape) and~~ where
the basal area of tree stands was high.

The diet and foraging habits of birds also moderated their response to
environmental variables (Fig. 3). Arboreal foragers were more abundant at sites with
greater FPC, more leaf litter, fewer miners and at higher elevations. Aerial foragers
were negatively associated with woody vegetation cover and FPC but preferred sites
with a simplified groundcover and with a more diverse composition of surrounding
landcover. Aerial foragers were also associated with woodlands that had a simplified
groundcover and a higher perimeter-area ratio. Birds with a mainly granivorous diet
were more common in woodlands at lower elevations and where FPC and woody
vegetation cover were low. In contrast, generalists and nectarivores were more
abundant with increasing cover of woody vegetation.

**Figure 3.** The relationship between birds that share species traits and environmental
 variables. Colours represent the strength of interactions (shading) and their direction
 (blue = negative & red = positive). The scale bar indicates the values of fourth-corner
 coefficients.

~~The diet and foraging habits of birds also moderated their response to~~
 ~~environmental variables (Fig. 3). Arboreal foragers were more abundant at sites with~~
 ~~greater FPC, more leaf litter and at higher elevations. Aerial foragers were negatively~~

~~associated with woody vegetation cover and preferred sites with a more diverse~~
~~composition of surrounding landcover. Aerial foragers were also associated with~~
~~woodlands that had a simplified groundcover and a higher perimeter-area ratio.~~
~~Birds with a mainly granivorous diet were more common in woodlands at lower~~
~~elevations and where FPC and woody vegetation cover were low. In contrast,~~
~~generalists and nectarivores were more abundant with increasing cover of woody~~
~~vegetation.~~

Our species distribution model indicates more complex relationships between
environmental variables and individual species (Fig. 4). For example, sulphur-crested
cockatoos (*Cacatua galerita*) were more likely to be found in areas of high woodland
cover but also preferred higher landcover diversity, higher groundcover diversity and
woodlands with less leaf litter. This apparently inconsistent relationship could reflect
the movement of cockatoos between large woodland sites where they were
frequently observed during the day and smaller satellite patches of degraded
woodland where they foraged in the morning and evening. Other parrot species (e.g.
blue-winged parrot *Neophema chrysostoma*, galah *Eolophus roseicapillus*, eastern
rosella) were also more common in areas with diverse landcover but small-bodied
and exotic birds (house sparrow, greenfinch and European goldfinch) had the
opposite response. Welcome swallows (*Hirundo neoxena*), swamp harriers (*Circus*
*approximans*) and tree martins (*Petrochelidon nigricans*) were positively associated
with shape and landcover diversity, which could reflect the preference of these
species to forage in edge habitat near to open water and native grasslands.

3.2 Models of change in species richness

Changes in species richness at historical survey sites were related to changes in
woodland cover and densities of noisy miners. Overall, though, there was generally
little change in native species richness (-1.48 ± 1.2 , range = 0 to 18) with nearly half
of all sites (15/33) gaining or losing fewer than two species (Fig. S1). The most
parsimonious models [$\Delta \text{AICc} < 7$ units, 52] explaining these changes always included
the effects of change in noisy miner density (Table 4). Sites where noisy miners had
increased in number were more likely to have experienced a decline in native
species richness. Change in woody vegetation cover was also a significant predictor:
sites that had a decline in surrounding woody vegetation also experienced a decline
in species richness. The initial size of the woodland patch had a smaller effect on
change in species richness, but larger patches tended to have fewer species
recorded than previously (Fig. S2). The best model included just these three
predictors and explained 45% (Adj. R^2) of the variation in the net change in species
richness. The top 3 models, however, were all within two ΔAICc units and carried
38%, 23% and 16% of the weight respectively (Table 4).

Only three survey sites experienced a reduction in patch size from clearing. Thus, our
analysis may not have had sufficient power to detect any effects of change in patch
size on species richness. The number of pivot irrigation systems increased
dramatically between 1997 ~~and 2017, when from there was a total of just~~ three
~~pivot irrigation systems~~irrigators within 1 km of two ~~of our~~ survey sites, ~~to and 2017~~
~~when there were~~ 33 irrigators ~~near to at~~ 12 survey sites, ~~but there~~
[revised manuscript text omitted]

**Competing Interests**

The authors declare no competing interests.

**Authors' Contribution**

GB collected the field data, helped conceive the study, performed the analysis and
wrote the paper; RG and RH helped collect the vegetation data and discussed the
results and statistical analyses; MM collected the historical field data and
participated in the design of the study; CJ and MJ helped to conceive and design the
study, discussed the results and statistical analyses and helped draft the manuscript.
All authors revised the manuscript and gave final approval for publication.

6. References

- 1 Emmerson, M., Morales, M. B., Oñate, J. J., Batáry, P., Berendse, F., Liira, J., Aavik,
819 T., Guerrero, I., Bommarco, R., Eggers, S., *et al.* 2016 How Agricultural Intensification
Affects Biodiversity and Ecosystem Services. In *Large-Scale Ecology: Model Systems*
*to Global Perspectives*. (ed. ^eds. A. J. Dumbrell, R. L. Kordas, G. Woodward), pp. 43-
97. London, UK.: Academic Press.
- Pisa, L. W., Amaral-Rogers, V., Belzunces, L. P., Bonmatin, J. M., Downs, C. A.,
Goulson, D., Kreutzweiser, D. P., Krupke, C., Liess, M., McField, M., *et al.* 2015 Effects
of neonicotinoids and fipronil on non-target invertebrates. *Environmental Science*
*and Pollution Research International*. **22**, 68-102. (10.1007/s11356-014-3471-x)
- Hallmann, C. A., Foppen, R. P., van Turnhout, C. A., de Kroon, H., Jongejans, E. 2014
Declines in insectivorous birds are associated with high neonicotinoid
concentrations. *Nature*. **511**, 341-343. (10.1038/nature13531)
- Boatman, N. D., Brickle, N. W., Hart, J. D., Milsom, T. P., Morris, A. J., Murray, A. W.
831 A., Murray, K. A., Robertson, P. A. 2004 Evidence for the indirect effects of pesticides
on farmland birds. *Ibis*. **146**, 131-143. (10.1111/j.1474-919X.2004.00347.x)
- Stanton, R. L., Morrissey, C. A., Clark, R. G. 2018 Analysis of trends and agricultural
drivers of farmland bird declines in North America: A review. *Agriculture, Ecosystems*
*& Environment*. **254**, 244-254. (10.1016/j.agee.2017.11.028)
- DeVault, T. L., Olson, Z. H., Beasley, J. C., Rhodes, O. E. 2011 Mesopredators
dominate competition for carrion in an agricultural landscape. *Basic and Applied*
*Ecology*. **12**, 268-274. (10.1016/j.baae.2011.02.008)
- Roos, S., Smart, J., Gibbons, D. W., Wilson, J. D. 2018 A review of predation as a
limiting factor for bird populations in mesopredator-rich landscapes: A case study of
the UK. *Biological Reviews of the Cambridge Philosophical Society*. **93**, 1915-1937.
(10.1111/brv.12426)

8 Lecq, S., Loisel, A., Brischoux, F., Mullin, S. J., Bonnet, X. 2017 Importance of
ground refuges for the biodiversity in agricultural hedgerows. *Ecol. Indic.* **72**, 615-
626. (10.1016/j.ecolind.2016.08.032)

9 Hunter, M. L., Acuña, V., Bauer, D. M., Bell, K. P., Calhoun, A. J. K., Felipe-Lucia, M.
R., Fitzsimons, J. A., González, E., Kinnison, M., Lindenmayer, D., *et al.* 2017
Conserving small natural features with large ecological roles: A synthetic overview.
*Biological Conservation.* **211**, 88-95. (10.1016/j.biocon.2016.12.020)

10 Denerley, C., Redpath, S. M., Wal, R., Newson, S. E., Chapman, J. W., Wilson, J. D.
2018 Breeding ground correlates of the distribution and decline of the common
cuckoo *Cuculus canorus* at two spatial scales. *Ibis.* **161**, 346-358. (10.1111/ibi.12612)

11 Fischer, J., Stott, J., Law, B. S. 2010 The disproportionate value of scattered trees.
*Biological Conservation.* **143**, 1564-1567. (10.1016/j.biocon.2010.03.030)

12 Fitzsimons, J. A., Michael, D. R. 2017 Rocky outcrops: A hard road in the
conservation of critical habitats. *Biological Conservation.* **211**, 36-44.
(10.1016/j.biocon.2016.11.019)

13 Hunter, M. C., Smith, R. G., Schipanski, M. E., Atwood, L. W., Mortensen, D. A.
2017 Agriculture in 2050: Recalibrating targets for sustainable intensification.
*BioScience.* **67**, 386-391. (10.1093/biosci/bix010)

14 Langlands, P. R., Brennan, K. E., Framenau, V. W., Main, B. Y. 2011 Predicting the
post-fire responses of animal assemblages: Testing a trait-based approach using
spiders. *J Anim Ecol.* **80**, 558-568. (10.1111/j.1365-2656.2010.01795.x)

15 Lindenmayer, D. B., Lane, P., Westgate, M., Scheele, B. C., Foster, C., Sato, C., Ikin,
865 K., Crane, M., Michael, D., Florance, D., *et al.* 2018 Tests of predictions associated
with temporal changes in Australian bird populations. *Biological Conservation.* **222**,
212-221. (10.1016/j.biocon.2018.04.007)

16 Howland, B. W. A., Stojanovic, D., Gordon, I. J., Radford, J., Manning, A. D.,
Lindenmayer, D. B. 2016 Birds of a feather flock together: Using trait-groups to

understand the effect of macropod grazing on birds in grassy habitats. *Biological*
*Conservation*. **194**, 89-99. (10.1016/j.biocon.2015.11.033)

17 Hanspach, J., Fischer, J., Ikin, K., Stott, J., Law, B. S. 2012 Using trait-based filtering
as a predictive framework for conservation: A case study of bats on farms in
southeastern Australia. *Journal of Applied Ecology*. **49**, 842-850. (10.1111/j.1365-
2664.2012.02159.x)

18 Bregman, T. P., Lees, A. C., MacGregor, H. E., Darski, B., de Moura, N. G., Aleixo,
877 A., Barlow, J., Tobias, J. A. 2016 Using avian functional traits to assess the impact of
878 land-cover change on ecosystem processes linked to resilience in tropical forests.
*Proceedings of the Royal Society B: Biological Sciences*. **283**,
(10.1098/rspb.2016.1289)

Ikin, K., Yong, D. L., Lindenmayer, D. B. 2016 Effectiveness of woodland birds as
taxonomic surrogates in conservation planning for biodiversity on farms. *Biological*
*Conservation*. **204**, 411-416. (10.1016/j.biocon.2016.11.010)

Wenny, D. G., DeVault, T. L., Johnson, M. D., Kelly, D., H. Sekercioglu, C.,
Tomback, D. F., Whelan, C. J. 2011 The Need to Quantify Ecosystem Services
Provided by Birds. *The Auk*. **128**, 1-14. (10.1525/auk.2011.10248)

Jones, M. E., Davidson, N. 2016 Applying an animal-centric approach to improve
ecological restoration. *Restoration Ecology*. **24**, 836-842. (10.1111/rec.12447)

Dingle, H. 2004 The Australo-Papuan bird migration system: Another
consequence of Wallace's Line. *Emu - Austral Ornithology*. **104**, 95-108.
(10.1071/mu03026)

Chan, K. 2001 Partial migration in Australian landbirds: A review. *Emu - Austral*
*Ornithology*. **101**, 281-292. (10.1071/mu00034)

Prior, L. D., Sanders, G. J., Bridle, K. L., Nichols, S. C., Harris, R., Bowman, D. M. J.
S. 2013 Land clearance not dieback continues to drive tree loss in a Tasmanian rural

landscape. *Regional Environmental Change*. **13**, 955-967. (10.1007/s10113-012-
0396-0)

MacDonald, M. A., Kirkpatrick, J. B. 2003 Explaining bird species composition and
richness in eucalypt-dominated remnants in subhumid Tasmania. *Journal of*
*Biogeography*. **30**, 1415-1426. (10.1046/j.1365-2699.2003.00927.x)

Maron, M., Grey, M. J., Catterall, C. P., Major, R. E., Oliver, D. L., Clarke, M. F.,
Loyn, R. H., Mac Nally, R., Davidson, I., Thomson, J. R., *et al.* 2013 Avifaunal disarray
due to a single despotic species. *Diversity and Distributions*. **19**, 1468-1479.
(10.1111/ddi.12128)

Department of the Environment and Energy. Listed Key Threatening Processes.
2019 [cited 2019 April 24]; Available from: [https://www.environment.gov.au/cgi-](https://www.environment.gov.au/cgi-bin/sprat/public/publicgetkeythreats.pl)
[bin/sprat/public/publicgetkeythreats.pl](https://www.environment.gov.au/cgi-bin/sprat/public/publicgetkeythreats.pl)

Hamer, R. 2019 Restoring farmland for biodiversity, a carnivorous perspective
[PhD]. Hobart, Tasmania: University of Tasmania.

Ford, H. A., Barrett, G. W., Saunders, D. A., Recher, H. F. 2001 Why have birds in
the woodlands of southern Australia declined? *Biological Conservation*. **97**, 71-88.
(10.1016/S0006-3207(00)00101-4)

BOM. Tasmania in 2016: A very wet and very warm year. 2018 [cited 2018 14
May]; Available from:
<http://www.bom.gov.au/climate/current/annual/tas/archive/2016.summary.shtml>

Bennett, J. C., Ling, F. L. N., Graham, B., Grose, M. R., Corney, S. P., White, C. J.,
Holz, G. K., Post, D. A., Gaynor, S. M., Bindoff, N. L. 2010 Climate Futures for
Tasmania: Water and Catchments Technical Report. Hobart, Tasmania: Antarctic
Climate & Ecosystems Cooperative Research Centre.

Loyn, R. H. 1986 The 20-minute search: A simple method for counting forest
birds. *Corella*. **10**, 58-60.

- Gardiner, R., Bain, G., Hamer, R., Jones, M. E., Johnson, C. N. 2018 Habitat
amount and quality, not patch size, determine persistence of a woodland-dependent
mammal in an agricultural landscape. *Landscape Ecology*. **33**, 1837–1849.
(10.1007/s10980-018-0722-0)
- Forman, R. T. T., Godron, M. 1986 *Landscape Ecology*. New York, USA: John Wiley
& Sons.
- ESRI. ArcGIS Desktop: Version 10.4.1. Redlands, CA, USA.: Environmental Systems
Research Institute 2019.
- DPIPWE. 2013 TASVEG 3.0. Hobart, Tasmania: Tasmanian Vegetation Monitoring
and Mapping Program, Resource Management and Conservation Division.
Department of Primary Industries, Parks, Water & the Environment.
- Gill, T., Johansen, K., Phinn, S., Trevithick, R., Scarth, P., Armston, J. 2016 A
method for mapping Australian woody vegetation cover by linking continental-scale
field data and long-term Landsat time series. *International Journal of Remote
Sensing*. **38**, 679-705. (10.1080/01431161.2016.1266112)
- Xu, T., Hutchinson, M. F. 2013 New developments and applications in the
ANUCLIM spatial climatic and bioclimatic modelling package. *Environmental
Modelling & Software*. **40**, 267-279. (10.1016/j.envsoft.2012.10.003)
- Thomson, J. R., Maron, M., Grey, M. J., Catterall, C. P., Major, R. E., Oliver, D. L.,
Clarke, M. F., Loyn, R. H., Davidson, I., Ingwersen, D., *et al.* 2015 Avifaunal disarray:
Quantifying models of the occurrence and ecological effects of a despotic bird
species. *Diversity and Distributions*. **21**, 451-464. (10.1111/ddi.12294)
- Marchant, S., Higgins, P. J. 1990 *Handbook of Australian, New Zealand and
Antarctic Birds. Volume 1: Rattites to Ducks*. Melbourne, Victoria: Oxford University
Press.

- Marchant, S., Higgins, P. J. 1993 *Handbook of Australian, New Zealand and*
*Antarctic Birds. Volume 2: Raptors to Lapwings*. Melbourne, Victoria: Oxford
University Press.
- Higgins, P. J., Davies, S. J. J. F. 1996 *Handbook of Australian, New Zealand and*
*Antarctic Birds. Volume 3: Snipe to Pigeons*. Melbourne, Victoria: Oxford University
Press.
- Higgins, P. J., Peter, J. M., Steele, W. K. 2001 *Handbook of Australian, New*
*Zealand and Antarctic Birds. Volume 5: Tyrant-flycatchers to Chats*. Melbourne,
Victoria: Oxford University Press.
- Higgins, P. J. 1999 *Handbook of Australian, New Zealand and Antarctic Birds.*
*Volume 4: Parrots to Dollarbird*. Melbourne, Victoria: Oxford University Press.
- Higgins, P. J., Peter, J. M. 2002 *Handbook of Australian, New Zealand and*
*Antarctic Birds. Volume 6: Pardalotes to Shrike-thrushes*. Melbourne, Victoria: Oxford
University Press.
- Higgins, P. J., Peter, J. M., Cowling, S. J. 2006 *Handbook of Australian, New*
*Zealand and Antarctic Birds. Volume 7: Boatbill to Starlings*. Melbourne, Victoria:
Oxford University Press.
- Hui, F. K. C., Poisot, T. 2016 boral - Bayesian ordination and regression analysis of
multivariate abundance data in R. *Methods in Ecology and Evolution*. **7**, 744-750.
(10.1111/2041-210x.12514)
- R Development Core Team. R: A language and environment for statistical
computing. R Foundation for Statistical Computing. 3.4.3 ed. Vienna, Austria 2017.
- Wang, Y., Naumann, U., Wright, S. T., Warton, D. I. 2012 mvabund- an R package
for model-based analysis of multivariate abundance data. *Methods in Ecology and*
*Evolution*. **3**, 471-474. (10.1111/j.2041-210X.2012.00190.x)

Warton, D. I., Shipley, B., Hastie, T., O'Hara, R. B. 2015 CATS regression - A model-
based approach to studying trait-based community assembly. *Methods in Ecology*
*and Evolution*. **6**, 389-398. (10.1111/2041-210x.12280)

Brown, A. M., Warton, D. I., Andrew, N. R., Binns, M., Cassis, G., Gibb, H., Yoccoz,
976 N. 2014 The fourth-corner solution - Using predictive models to understand how
species traits interact with the environment. *Methods in Ecology and Evolution*. **5**,
344-352. (10.1111/2041-210x.12163)

Burnham, K. P., Anderson, D. R., Huyvaert, K. P. 2010 AIC model selection and
multimodel inference in behavioral ecology: some background, observations, and
comparisons. *Behavioral Ecology and Sociobiology*. **65**, 23-35. (10.1007/s00265-010-
1029-6)

Dow, D. D. 1977 Indiscriminate interspecific aggression leading to almost sole
occupancy of space by a single species of bird. *Emu - Austral Ornithology*. **77**, 115-
121. (10.1071/MU9770115)

Cunningham, R. B., Lindenmayer, D. B., Crane, M., Michael, D. R., Barton, P. S.,
Gibbons, P., Okada, S., Ikin, K., Stein, J. A. R., Heikkinen, R. 2014 The law of
diminishing returns: woodland birds respond to native vegetation cover at multiple
spatial scales and over time. *Diversity and Distributions*. **20**, 59-71.
(10.1111/ddi.12145)

Villard, M. A., Trzcinski, M. K., Merriam, G. 1999 Fragmentation effects on forest
birds: Relative influence of woodland cover and configuration on landscape
occupancy. *Conservation Biology*. **13**, 774-783. (DOI 10.1046/j.1523-
1739.1999.98059.x)

Heikkinen, R. K., Luoto, M., Virkkala, R., Rainio, K. 2004 Effects of habitat cover,
landscape structure and spatial variables on the abundance of birds in an
agricultural-forest mosaic. *Journal of Applied Ecology*. **41**, 824-835. (10.1111/j.0021-
8901.2004.00938.x)

Smith, A. C., Fahrig, L., Francis, C. M. 2011 Landscape size affects the relative
importance of habitat amount, habitat fragmentation, and matrix quality on forest
birds. *Ecography*. **34**, 103-113. (10.1111/j.1600-0587.2010.06201.x)

Ikin, K., Barton, P. S., Stirnemann, I. A., Stein, J. R., Michael, D., Crane, M., Okada,
S., Lindenmayer, D. B. 2014 Multi-scale associations between vegetation cover and
woodland bird communities across a large agricultural region. *PLoS One*. **9**, e97029.
(10.1371/journal.pone.0097029)

Watson, J. E. M., Whittaker, R. J., Freudenberger, D. 2005 Bird community
responses to habitat fragmentation: How consistent are they across landscapes?
*Journal of Biogeography*. **32**, 1353-1370. (10.1111/j.1365-2699.2005.01256.x)

Connor, E. F., Courtney, A. C., Yoder, J. M. 2000 Individuals–area relationships:
The relationship between animal population density and area. *Ecology*. **81**, 734-748.
(10.1890/0012-9658(2000)081[0734:lartrb]2.0.Co;2)

Zarette, L., Doyle, P., Trémont, S. M. 2000 Food shortage in small fragments:
Evidence from an area-sensitive passerine. *Ecology*. **81**, 1654-1666. (10.1890/0012-
9658(2000)081[1654:fsisfe]2.0.co;2)

Hartley, M. J., Hunter, M. L. 1998 A meta-analysis of forest cover, edge effects,
and artificial nest predation rates. *Conservation Biology*. **12**, 465-469.
(10.1111/j.1523-1739.1998.96373.x)

Fletcher, R. J. 2009 Does attraction to conspecifics explain the patch-size effect?
An experimental test. *Oikos*. **118**, 1139-1147. (10.1111/j.1600-0706.2009.17342.x)

Murcia, C. 1995 Edge effects in fragmented forests - Implications for
conservation. *Trends Ecol. Evol.* **10**, 58-62. (10.1016/s0169-5347(00)88977-6)

Bélisle, M., Desrochers, A., Fortin, M.-J. 2001 Influence of forest cover on the
movements of forest birds: A homing experiment. *Ecology*. **82**, 1893-1904.
(10.1890/0012-9658(2001)082[1893:lofcot]2.0.Co;2)

- O'Loughlin, T., O'Loughlin, L. S., Clarke, M. F. 2017 No short-term change in avian
assemblage following removal of yellow-throated miner (*Manorina flavigula*)
colonies. *Ecological Management & Restoration*. **18**, 83-87. (10.1111/emr.12244)
- Kutt, A. S., Vanderduys, E. P., Perry, J. J., Mathieson, M. T., Eyre, T. J. 2016 Yellow-
throated miners *Manorina flavigula* homogenize bird communities across intact and
fragmented landscapes. *Austral Ecology*. **41**, 316-327. (10.1111/aec.12314)
- Davitt, G., Maute, K., Major, R. E., McDonald, P. G., Maron, M. 2018 Short-term
response of a declining woodland bird assemblage to the removal of a despotic
competitor. *Ecol. Evol.* **8**, 4771-4780. (10.1002/ece3.4016)
- Montague-Drake, R. M., Lindenmayer, D. B., Cunningham, R. B. 2009 Factors
affecting site occupancy by woodland bird species of conservation concern.
*Biological Conservation*. **142**, 2896-2903. (10.1016/j.biocon.2009.07.009)
- Taylor, S. G. 2008 Leaf litter invertebrate assemblages in box-ironbark forest:
Compostion, size and seasonal variation in biomass. *The Victorian Naturalist*. **125**,
19-27.
- Watson, D. M. 2011 A productivity-based explanation for woodland bird declines:
Poorer soils yield less food. *Emu*. **111**, 10-18. (10.1071/mu09109)
- McGregor, H., Legge, S., Jones, M. E., Johnson, C. N. 2015 Feral cats are better
killers in open habitats, revealed by animal-borne video. *PLoS One*. **10**, e0133915.
(10.1371/journal.pone.0133915)
- Ford, J. 1981 Evolution, distribution and stage of speciation in the *Rhipidura*
*fuliginosa* complex in Australia. *Emu - Austral Ornithology*. **81**, 128-144.
- Radford, J. Q., Bennett, A. F. 2007 The relative importance of landscape
properties for woodland birds in agricultural environments. *Journal of Applied*
*Ecology*. **44**, 737-747. (10.1111/j.1365-2664.2007.01327.x)

MacDonald, M. A. 2001 Habitat fragmentation and its effects on birds and
grasshoppers in eucalypt remnants in the tasmanian midlands. Hobart, Tasmania:
University of Tasmania.

Green, R. H. 1984 Little corella in Tasmania. *The Tasmanian Naturalist*. **77**, 6.

Cunningham, C. X., Johnson, C. N., Barmuta, L. A., Hollings, T., Woehler, E. J.,
Jones, M. E. 2018 Top carnivore decline has cascading effects on scavengers and
carrion persistence. *proceedings of the Royal Society B: Biological Sciences*. **285**,
(10.1098/rspb.2018.1582)

Montague-Drake, R. M., Lindenmayer, D. B., Cunningham, R. B., Stein, J. A. 2011 A
reverse keystone species affects the landscape distribution of woodland avifauna: A
case study using the noisy miner (*Manorina melanocephala*) and other Australian
birds. *Landscape Ecology*. **26**, 1383-1394. (10.1007/s10980-011-9665-4)

Clarke, M. F., Oldland, J. M. 2007 Penetration of remnant edges by noisy miners
(*Manorina melanocephala*) and implications for habitat restoration. *Wildl. Res.* **34**,
253-261. (10.1071/wr06134)

Robertson, O., Maron, M., Buckley, Y., McAlpine, C. 2013 Incidence of
competitors and landscape structure as predictors of woodland-dependent birds.
*Landscape Ecology*. **28**, 1975-1987. (10.1007/s10980-013-9934-5)

Hastings, R. A., Beattie, A. J. 2006 Stop the bullying in the corridors: Can including
shrubs make your revegetation more noisy miner free? *Ecological Management &*
*Restoration*. **7**, 105-112.

Taylor, R. S., Oldland, J. M., Clarke, M. F. 2008 Edge geometry influences patch-
level habitat use by an edge specialist in south-eastern Australia. *Landscape Ecology*.
**23**, 377-389. (10.1007/s10980-008-9196-9)

Marzluff, J. M., Ewing, K. 2001 Restoration of fragmented landscapes for the
conservation of birds: A general framework and specific recommendations for

urbanizing landscapes. *Restoration Ecology*. **9**, 280-292. (DOI 10.1046/j.1526-
100x.2001.009003280.x)

Spooner, P. G. 2005 Response of *Acacia* species to disturbance by roadworks in
roadside environments in southern New South Wales, Australia. *Biological*
*Conservation*. **122**, 231-242. (10.1016/j.biocon.2004.07.012)

Fischer, J., Lindenmayer, D. B. 2002 The conservation value of paddock trees for
birds in a variegated landscape in southern New South Wales; Paddock trees as
stepping stones. *Biodiversity and Conservation*. **11**, 807-832.
(10.1023/a:1015371511169)

Major, R. E., Christie, F. J., Gowing, G. 2001 Influence of remnant and landscape
attributes on Australian woodland bird communities. *Biological Conservation*. **102**,
47-66. (10.1016/s0006-3207(01)00090-8)

Grey, M. J., Clarke, M. F., Loyn, R. H. 1997 Initial changes in the avian
communities of remnant eucalypt woodlands following a reduction in the
abundance of noisy miners, *Manorina melanocephala*. *Wildl. Res.* **24**, 631-648.
(10.1071/WR96080)

Debus, S. J. S. 2008 The effect of noisy miners on small bush birds - An unofficial
cull and its outcome. *Pacific Conservation Biology*. **14**, 185-190.

Crates, R., Terauds, A., Rayner, L., Stojanovic, D., Heinsohn, R., Wilkie, C., Webb,
1095 M. 2018 Spatially and temporally targeted suppression of despotic noisy miners has
1096 conservation benefits for highly mobile and threatened woodland birds. *Biological*
*Conservation*. **227**, 343-351. (10.1016/j.biocon.2018.10.006)

Law, B. S., Chidel, M., Brassil, T., Turner, G., Kathuria, A. 2014 Trends in bird
diversity over 12 years in response to large-scale eucalypt plantation establishment:
Implications for extensive carbon plantings. *Forest Ecology and Management*. **322**,
58-68.

- Huth, N., Possingham, H. P. 2011 Basic ecological theory can inform habitat
restoration for woodland birds. *Journal of Applied Ecology*. **48**, 293-300.
(10.1111/j.1365-2664.2010.01936.x)
- Hobbs, R., Catling, P. C., Wombey, J. C., Clayton, M., Atkins, L., Reid, A. 2003
Faunal use of bluegum (*Eucalyptus globulus*) plantations in southwestern Australia.
*Agroforestry Systems*. **58**, 195-212. (10.1023/A:1026073906512)
- Kavanagh, R. P., Stanton, M. A., Herring, M. W. 2007 Eucalypt plantings on farms
benefit woodland birds in south-eastern Australia. *Austral Ecology*. **32**, 635-650.
(10.1111/j.1442-9993.2007.01746.x)
- Loyn, R., McNabb, E., Macak, P., Noble, P. 2007 Eucalypt plantations as habitat for
birds on previously cleared farmland in south-eastern Australia. *Biological*
*Conservation*. **137**, 533-548. (10.1016/j.biocon.2007.03.012)
- Sogge, M. K., Sferra, S. J., Paxton, E. H. 2008 Tamarix as habitat for birds:
Implications for riparian restoration in the southwestern United States. *Restoration*
*Ecology*. **16**, 146-154. (10.1111/j.1526-100X.2008.00357.x)
- Koch, A. J., Munks, S. A., Woehler, E. J. 2008 Hollow-using vertebrate fauna of
Tasmania: Distribution, hollow requirements and conservation status. *Australian*
*Journal of Zoology*. **56**, 323-349. (10.1071/Zo08003)
- Koch, A. J., Munks, S. A., Driscoll, D., Kirkpatrick, J. B. 2008 Does hollow
occurrence vary with forest type? A case study in wet and dry *Eucalyptus obliqua*
forest. *Forest Ecology and Management*. **255**, 3938-3951.
(10.1016/j.foreco.2008.03.025)
- Le Roux, D. S., Ikin, K., Lindenmayer, D. B., Bistricher, G., Manning, A. D., Gibbons,
P. 2016 Enriching small trees with artificial nest boxes cannot mimic the value of
large trees for hollow-nesting birds. *Restoration Ecology*. **24**, 252-258.
(10.1111/rec.12303)

- Inger, R., Gregory, R., Duffy, J. P., Stott, I., Vorisek, P., Gaston, K. J. 2015 Common
European birds are declining rapidly while less abundant species' numbers are rising.
*Ecology Letters*. **18**, 28-36. (10.1111/ele.12387)
- Radford, J. Q., Bennett, A. F., Cheers, G. J. 2005 Landscape-level thresholds of
habitat cover for woodland-dependent birds. *Biological Conservation*. **124**, 317-337.
(10.1016/j.biocon.2005.01.039)

Appendix D

Professor Kevin Padian - Subject Editor (Biology)
University of California, Berkley
Department of Integrative Biology
Berkley, California.

11th February 2020

Dear Professor Padian,

As advised by the Editorial Coordinator, Anita Kristiansen, I have revised the order and content of the end statements for final submission. I understand that no further comments need to be addressed regarding the paper itself.

Many thanks,

Glen Bain

School of Natural Sciences
Biological Sciences | University of Tasmania
Email: glen.bain@utas.edu.au
Phone: +61 405063230